# An end-to-end attention-based approach for learning on graphs

David Buterez [1] ✉, Jon Paul Janet [2], Dino Oglic[3] & Pietro Liò [1]

There has been a recent surge in transformer-based architectures for learning on graphs, mainly motivated by attention as an effective learning mechanism and the desire to supersede the hand-crafted operators characteristic of message passing schemes. However, concerns over their empirical effectiveness, scalability, and complexity of the pre-processing steps have been raised, especially in relation to much simpler graph neural networks that typically perform on par with them across a wide range of benchmarks. To address these shortcomings, we consider graphs as sets of edges and propose a purely attention-based approach consisting of an encoder and an attention pooling mechanism. The encoder vertically interleaves masked and vanilla self-attention modules to learn an effective representation of edges while allowing for tackling possible misspecifications in input graphs. Despite its simplicity, the approach outperforms fine-tuned message passing baselines and recently proposed transformer-based methods on more than 70 node and graph-level tasks, including challenging long-range benchmarks. Moreover, we demonstrate state-of-the-art performance across different tasks, ranging from molecular to vision graphs, and heterophilous node classification. The approach also outperforms graph neural networks and transformers in transfer learning settings and scales much better than alternatives with a similar performance level or expressive power.

We empirically investigate the potential of a purely attention-based approach to learn effective representations of graph-structured data. Typically, learning on graphs is modelled as message passing, an iterative process that relies on a message function to aggregate information from a given node's neighbourhood and an update function to incorporate the encoded message into the output representation of the node. The resulting graph neural networks (GNNs) typically stack multiple such layers to learn node representations based on vertex-rooted subtrees, essentially mimicking the one-dimensional Weisfeiler–Lehman (1-WL) graph isomorphism test[1,2]. Variations of message passing have been applied effectively in different fields such as life sciences[3–9], electrical engineering[10], and weather prediction[11].

Despite the overall success and wide adoption of graph neural networks, several practical challenges have been identified over time.

Although the message passing framework is highly flexible, the design of new layers is a challenging research problem where improvements take years to achieve and often rely on hand-crafted operators. This is particularly the case for general-purpose graph neural networks that do not exploit additional input modalities, such as atomic coordinates. For example, principal neighbourhood aggregation (PNA) is regarded as one of the most powerful message passing layers[12], but it is built using a collection of manually selected neighbourhood aggregation functions, requires a degree histogram of the dataset which must be precomputed prior to learning, and further uses manually selected degree scaling. The nature of message passing also imposes certain limitations that have shaped the majority of the literature. One of the most prominent examples is the readout function used to combine node-level features into a single graph-level representation, which is

[1]Department of Computer Science and Technology, University of Cambridge, Cambridge, UK. [2]Molecular AI, BioPharmaceuticals R&D, AstraZeneca, Gothenburg, Sweden. [3]Centre for AI, BioPharmaceuticals R&D, AstraZeneca, Cambridge, UK. ✉e-mail: db804@cantab.ac.uk

required to be permutation invariant with respect to the node order. Thus, the default choice for graph neural networks and even graph transformers remains a simple, non-learnable function such as sum, mean, or max[13–15]. The limitations of this approach have been identified by Wagstaff et al.[16], who have shown that simple readout functions might require complex item embedding functions that are difficult to learn using standard neural networks. Additionally, graph neural networks have shown limitations in terms of over-smoothing[17–19], linked to node representations becoming similar with increased depth, and over-squashing[20,21] due to information compression through bottleneck edges. The former has been associated with poor performance on node classification tasks with heterophilic graphs, and it is hypothesised that this is due to GNNs acting as low-pass filters. Recently, Di Giovanni et al.[18] have studied over-smoothing using gradient flows on graphs and have demonstrated that some time-continuous GNNs are indeed dominated by low frequencies. Moreover, behaviour opposite to over-smoothing, known as over-sharpening, has been identified in a setting with linear graph convolutions and symmetric weights, achieved via repulsion induced by negative eigenvalues of the weight matrix. Over-squashing, on the other hand, impacts performance in cases where information from distant nodes is relevant for predictive tasks, and it has been linked to graph bottlenecks, characterised by a rapid increase in the number of $k$-hop neighbourhoods as $k$ or the number of layers increases. Topping et al.[21] have provided a theoretical characterisation of over-squashing and introduced a notion of graph curvature as means for quantifying it, along with a graph rewiring algorithm known as stochastic discrete Ricci flow that can mitigate some of the bottleneck effects. Alternative solutions for these two problems typically take the form of message regularisation schemes[22–24]. There is, however, no clear consensus on the right architectural choices for building effective deep message passing neural networks and effectively addressing all of those challenges. Transfer learning and strategies such as pre-training and fine-tuning are also less ubiquitous in graph neural networks because of modest or ambiguous benefits, as opposed to large language models[25].

The attention mechanism[26] is one of the main sources of innovation within graph learning, either by directly incorporating attention within message passing[27,28], by formulating graph learning as a language processing task[29,30], or by combining vanilla GNN layers with attention layers[13,14,31–33]. However, several concerns have been raised regarding the performance, scalability, and complexity of such methods. Performance-wise, recent reports indicate that sophisticated graph transformers underperform compared to simple but tuned GNNs[34,35]. This line of work highlights the importance of empirically evaluating new methods relative to strong baselines. Separately, recent graph transformers have focused on increasingly more complex helper mechanisms, such as computationally expensive pre-processing and learning steps[29], various different encodings (e.g. positional, structural, and relational)[29–31,36], inclusion of virtual nodes and edges[29,32], conversion of the problem to natural language processing[29,30], and other non-trivial graph transformations[32,36]. These complications can significantly increase computational requirements, reducing the chance of being widely adopted and replacing GNNs.

Motivated by the effectiveness of attention as a learning mechanism and recent advances in efficient and exact attention, we introduce an end-to-end attention-based architecture for learning on graphs that is simple to implement, scalable, and achieves state-of-the-art results. The proposed architecture considers graphs as sets of edges, leveraging an encoder that interleaves masked and self-attention mechanisms to learn effective representations. The attention-based pooling component mimics the functionality of a readout function and is responsible for aggregating the edge-level features into a permutation-invariant graph-level representation. The masked attention mechanism allows for learning effective edge representations originating from the graph connectivity, and the

combination with self-attention layers vertically allows for expanding on this information while having a strong prior. Masking can thus be seen as leveraging specified relational information and its vertical combination with self-attention as a means to overcome possible misspecification of the input graph. The masking operator is injected into the pairwise attention weight matrix and allows only for attention between linked primitives. For a pair of edges, connectivity translates to having a shared node between them. We focus primarily on learning through edge sets due to empirically high performance and refer to our architecture as edge-set attention (ESA). We also show that the overall architecture is effective in propagating information across nodes through dedicated node-level benchmarks. The ESA architecture is general purpose, in the sense that it relies only on the graph structure and possibly node and edge features, and it is not restricted to any particular domain. Furthermore, ESA does not rely on positional, structural, relative, or similar encodings, it does not encode graph structures as tokens or other language (sequence) specific concepts, and it does not require any pre-computations.

Despite its apparent simplicity, ESA-based learning overwhelmingly outperforms strong and tuned GNN baselines and much more involved transformer-based models. Our evaluation is extensive, totalling 70 datasets and benchmarks from different domains such as quantum mechanics, molecular docking, physical chemistry, biophysics, bioinformatics, computer vision, social networks, functional call graphs, and synthetic graphs. At the node level, we include both homophilous and heterophilous graph tasks, as well as shortest path problems, as these require modelling long-range interactions. Beyond supervised learning tasks, we explore the potential for transfer learning in the context of drug discovery and quantum mechanics[37,38] and show that ESA is a viable transfer learning strategy compared to vanilla GNNs and graph transformers.

## Related work

An attention mechanism that mimics message passing and that limits the attention computations to neighbouring nodes was first proposed in GAT[27]. We consider masking as an abstraction of the GAT attention operator that allows for building general-purpose relational structures between items in a set (e.g. $k$-hop neighbourhoods or conformational masks extracted from 3D molecular structures). The attention mechanism in GAT is implemented as a single linear projection matrix that does not explicitly distinguish between keys, queries, and values as in standard dot product attention and an additional linear layer after concatenating the representations of connected nodes, along with a non-linearity. This type of simplified attention has been labelled in subsequent work as static and was shown to have limited expressive power[28]. Brody et al. instead proposed dynamic attention, a simple reordering of operations in GAT, resulting in a more expressive GATv2 model[28]—however, at the price of doubling the parameter count and the corresponding memory consumption. A high-level overview of GAT in the context of masked attention is provided in Fig. 1A, B, along with the main differences to the proposed architecture. Similarly to GAT, several adaptations of the original scaled dot product attention have been proposed for graphs[39,40], where the focus was on defining an attention mechanism constrained by node connectivity and replacing the positional encodings of the original transformer model with more appropriate graph alternatives, such as Laplacian eigenvectors. These approaches, while interesting and forward-looking, did not convincingly outperform simple GNNs[39,40]. Building on this line of work, an architecture that can be seen as an instance of masked transformers has been proposed in SAN[41], illustrated in Fig. 1C, D. The attention coefficients are defined as a convex combination of the scores (controlled by hyperparameter $\gamma$) associated with the original graph and its complement. Min et al.[42] have also considered a masking mechanism for standard transformers. However, the graph structure itself is not

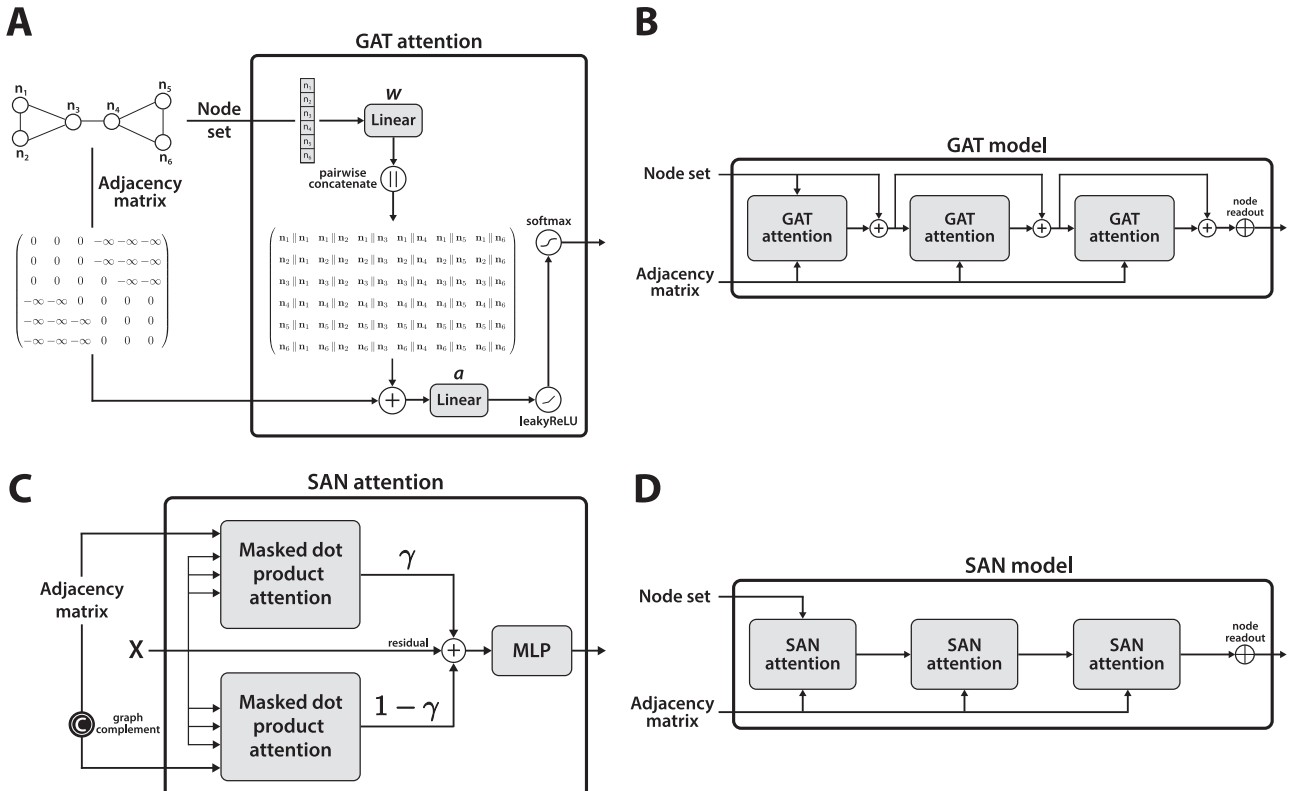

**Fig. 1 | The GAT and SAN architectures presented schematically. A**, **B**. A high-level overview of the GAT message passing algorithm[27]. **A** A GAT layer receives node representations and the adjacency matrix as inputs. First, a projection matrix is applied to all the nodes, which are then concatenated pairwise and passed through another linear layer, followed by a non-linearity. The final attention score is computed by the softmax function. **B** A GAT model stacks multiple GAT layers and uses a readout function over nodes to generate a graph-level representation. Residual connections are also illustrated as a modern enhancement of GNNs (not included in the original approach). **C**, **D**. A high-level overview of the SAN architecture[41]. **C** Two independent attention modules with a shared value projection matrix are used for the input graph and its complement. The outputs of these two modules are convexly combined using a hyperparameter $\gamma$ before being residually added to the input. **D** The overall SAN architecture that stacks multiple SAN layers without residual connections and uses a standard readout function over nodes. The differences between ESA, GAT, and SAN are detailed in SI 1.

used directly in the masking process as the devised masks correspond to four types of prior interaction graphs (induced, similarity, cross neighbourhood, and complete subgraphs), acting as an inductive bias. Furthermore, the method was not designed for general-purpose graph learning as it relies on helper mechanisms such as neighbour sampling and a heterogeneous information network.

Recent trends in learning on graphs are dominated by architectures based on standard self-attention layers as the only learning mechanism, with a significant amount of effort put into representing the graph structure exclusively through positional and structural encodings. Graphormer[29] is one of the most prominent approaches of this class. It comes with an involved and computation-heavy suite of pre-processing steps, involving a centrality, spatial, and edge encoding. For instance, spatial encodings rely on the Floyd–Warshall algorithm, which has cubic time complexity in the number of nodes and quadratic memory complexity. In addition, the model employs a virtual mean readout node that is connected to all other nodes in the graph. Although Graphormer was originally evaluated on only four datasets, the results were promising relative to GNNs and SAN. Another related approach is the Tokenized Graph Transformer (TokenGT)[30], which treats all nodes and edges as independent tokens. To adapt sequence learning to the graph domain, TokenGT encodes the graph information using node identifiers derived from orthogonal random features or Laplacian eigenvectors, and learnable type identifiers for nodes and edges. TokenGT is provably more expressive than standard GNNs and can approximate $k$-WL tests with the appropriate architecture (number of layers, adequate pooling, etc.). However, this theoretical guarantee holds only with the use of positional encodings that typically break the permutation invariance over nodes that is required for consistent predictions over the same graph presented with a different node order. The main strength of TokenGT is its theoretical expressiveness, as it has only been evaluated on a single dataset where it did not outperform Graphormer.

A route involving a hybrid between classical transformers and message passing has been pursued in the GraphGPS framework[43], which combines message passing and transformer layers. As in previous works, GraphGPS places a large emphasis on different types of encodings, proposing and analysing positional and structural encodings, further divided into local, global, and relative encodings. Exphormer[32] is an evolution of GraphGPS that adds virtual global nodes and sparse attention based on expander graphs. While effective, such frameworks still rely on message passing and are thus not purely attention-based solutions. Limitations include dependence on approximations (Performer[44] for both approaches, expander graphs for Exphormer), and decreased performance when encodings (GraphGPS) or special nodes (Exphormer) are removed. Notably, Exphormer is the first approach from this class to consider custom attention patterns given by node neighbourhoods. Polynormer[33] is another approach from this class that combines local (mimicking graph convolutions or GATs) and global attention (scaled dot product attention), and achieves polynomial expressivity by a gating mechanism which can be seen as an alternative to residual connections, with element-wise multiplication between the projected inputs and outputs of the attention blocks.

**Algorithm 1.** Edge masking in PyTorch Geometric (T is the transpose; helper functions are explained in SI 8).

```
1   from esa import consecutive, first_unique_index
2   function edge_adjacency(batched_edge_index)
3       N_e ← batched_edge_index.size(1)
4       source_nodes ← batched_edge_index[0]
5       target_nodes ← batched_edge_index[1]
6
7       # unsqueeze and expand
8       exp_src ← source_nodes.unsq(1).expand((-1, N_e))
9       exp_trg ← target_nodes.unsq(1).expand((-1, N_e))
10
11      src_adj ←  exp_src == T(exp_src)
12      trg_adj ←  exp_trg == T(exp_trg)
13        cross ← (exp_src == T(exp_trg)) logical_or
14                (exp_trg == T(exp_src))
15
16      return (src_adj logical_or trg_adj logical_or cross)
```

```
17  function edge_mask(b_ei, b_map, B, L)
18      mask ← torch.full(size=(B, L, L), fill=False)
19      edge_to_graph ← b_map.index_select(0, b_ei[0, :])
20
21      edge_adj ← edge_adjacency(b_ei)
22      ei_to_original ← consecutive(
23          first_unique_index(edge_to_graph), b_ei.size(1))
24
25      edges ← edge_adj.nonzero()
26      graph_index ← edge_to_graph.idx_select(0, edges[:, 0])
27      coord_1 ← ei_to_original.idx_select(0, edges[:, 0])
28      coord_2 ← ei_to_original.idx_select(0, edges[:, 1])
29
30      mask[graph_index, coord_1, coord_2] ← True
31      return ~mask
```

# Results

We perform a comprehensive evaluation of ESA on 70 different tasks, including domains such as molecular property prediction, vision graphs, and social networks, as well as different aspects of representation learning on graphs, ranging from node-level tasks with homophily and heterophily graph types to modelling long range dependencies, shortest paths, and 3D atomic systems. We quantify the performance of our approach relative to 6 GNN baselines: GCN, GAT, GATv2, PNA, GIN, and DropGIN (more expressive than 1-WL), and 3 graph transformer baselines: Graphormer, TokenGT, and GraphGPS. All the details on hyperparameter tuning, rationale for the selected metrics, and selection of baselines can be found in SI 9.1 to 9.3. In the remainder of the section, we summarise our findings across molecular learning, mixed graph-level tasks, node-level tasks, ablations on the interleaving operator, along with insights on time and memory scaling, and explainability enabled by the learnt attention weights.

## Molecular learning

As learning on molecules has emerged as one of the most successful applications of graph learning, we present an in-depth evaluation that includes quantum mechanics, molecular docking, and various physical chemistry and biophysics benchmarks, as well as an exploration of learning on 3D atomic systems, transfer learning, and learning on large molecules of therapeutic relevance (peptides).

**QM9.** We report results for all 19 QM9[45] targets in Table 1, with GCN and GIN separately in Supplementary Table 1 due to space restrictions. We observe that on 15 out of 19 properties, ESA is the best-performing model. The exceptions are the frontier orbital energies (HOMO and LUMO energy, HOMO-LUMO gap) and the dipole moment ($\mu$), where PNA is slightly ahead of it. Other graph transformers are competitive relative to GNNs on many properties, but vary in performance across tasks.

**DOCKSTRING.** DOCKSTRING[46] is a recent drug discovery data collection consisting of molecular docking scores for 260,155 small molecules and 5 high-quality targets from different protein families that were selected as a regression benchmark, with different levels of difficulty: PARP1 (enzyme, easy), F2 (protease, easy to medium), KIT (kinase, medium), ESR2 (nuclear receptor, hard), and PGR (nuclear receptor, hard). We report results for the 5 targets in Table 1 (and Supplementary Table 1) and observe that ESA is the best performing method on 4 out of 5 tasks, with PNA slightly ahead on the medium-difficulty KIT. TokenGT and GraphGPS generally do not match ESA or even PNA. Molecular docking scores also depend heavily on 3D geometry, as discussed in the original paper[46], posing a difficult challenge for all methods. Interestingly, not only ESA but all tuned GNN baselines outperform the strongest method in the original manuscript (Attentive FP, a GNN based on attention[47]) despite using 20,000 fewer training

molecules (which we leave out as a validation set). This illustrates the importance of evaluating relative to baselines with tuned hyperparameters.

**MoleculeNet and NCI.** We report results for a varied selection of three regression and three classification benchmarks from MoleculeNet, as well as two benchmarks from the National Cancer Institute (NCI), consisting of compounds screened for anti-cancer activity (Tables 1 and 3, and Supplementary Tables 1 and 5). We also report the accuracy in Supplementary Table 6. Apart from HIV, these datasets pose a challenge to graph transformers due to their small size (<5000 compounds). With the exception of the Lipophilicity (LIPO) dataset from MoleculeNet, we observe that ESA is the preferred method, and often by a significant margin, for example on BBBP and BACE, despite their small size (2039, respectively 1513 total compounds before splitting). On the other hand, Graphormer and TokenGT perform poorly, possibly due to the small-sample nature of these tasks. GraphGPS is closer to the top performers, but still underperforms compared to GAT(v2) and PNA. These results also show the importance of appropriate evaluation metrics, as the accuracy on HIV for all methods is above 97% (Supplementary Table 6), but the MCC (Table 3) is significantly lower.

**PCQM4MV2.** PCQM4MV2 is a quantum chemistry benchmark introduced as a competition through the Open Graph Benchmark Large-Scale Challenge (OGB-LSC) project[48]. It consists of 3,378,606 training molecules with the goal of predicting the DFT-calculated HOMO-LUMO energy gap from 2D molecular graphs. It has been widely used in the literature, especially to evaluate graph transformers[29,30,49]. Since the test splits are not public, we use the available validation set (73,545 molecules) as a test set, as is often done in the literature, and report results on it after training for 400 epochs. We report results from a single run, as it is common in the field due to the large size of the dataset[29,30,49]. For the same reason, we do not run our baselines and instead choose to focus on the state-of-the-art results published on the official leaderboard. At the time of writing (March 2025), the best-performing model achieved a validation set MAE of 0.0671[50]. Here, ESA achieves a value of 0.0235, which is almost 3 times lower. It is worth noting that the top 3 methods for this dataset are all bespoke architectures designed for molecular learning, for example, Uni-Mol+[51], Transformer-M[52], and TGT[50]. In contrast, ESA is general-purpose (does not use any information or technique specifically for molecular learning), uses only the 2D input graph, and does not use any positional or structural encodings.

**ZINC.** We report results on the full ZINC dataset with 250,000 compounds (Table 1), which is commonly used for generative purposes[53,54]. This is one of the only benchmarks where the graph transformer baselines (Graphormer, TokenGT, GraphGPS) convincingly

**Table 1 | The table reports the root mean squared error (RMSE) on QM9, the mean absolute error (MAE) for ZINC, and $R^2$ for DOCKSTRING (DOCK) and MoleculeNet, presented as mean ± standard deviation over 5 runs**

| | Target | DropGIN | GAT | GATv2 | PNA | Graphormer | TokenGT | GPS | ESA |
|---|---|---|---|---|---|---|---|---|---|
| QM9 (↓) | $\mu$ | 0.55±0.01 | 0.55±0.01 | 0.55±0.01 | **0.53±0.01** | 0.63±0.01 | 0.76±0.02 | 0.94±0.17 | 0.56±0.00 |
| | $\alpha$ | 0.44±0.03 | 0.48±0.02 | 0.46±0.02 | 0.48±0.07 | 0.40±0.02 | 0.45±0.01 | 0.60±0.13 | **0.40±0.00** |
| | $\varepsilon_{HOMO}$ | 0.11±0.00 | 0.11±0.00 | 0.10±0.00 | **0.10±0.00** | 0.11±0.00 | 0.12±0.00 | 0.11±0.00 | 0.10±0.00 |
| | $\varepsilon_{LUMO}$ | 0.12±0.00 | 0.12±0.00 | 0.13±0.00 | **0.11±0.00** | 0.11±0.00 | 0.13±0.00 | 0.11±0.00 | 0.11±0.00 |
| | $\Delta\varepsilon$ | 0.17±0.00 | 0.16±0.01 | 0.16±0.00 | **0.14±0.00** | 0.16±0.00 | 0.18±0.01 | 0.15±0.01 | 0.15±0.00 |
| | $\langle R^2 \rangle$ | 29.87±0.34 | 30.91±0.15 | 30.15±0.36 | 28.50±0.44 | 29.63±0.35 | 31.54±0.42 | 30.42±1.19 | **28.33±0.32** |
| | ZPVE | 0.03±0.00 | 0.03±0.00 | 0.03±0.00 | 0.03±0.00 | 0.06±0.03 | 0.03±0.00 | 0.03±0.01 | **0.03±0.00** |
| | $U_0$ | 29.82±6.32 | 27.82±3.98 | 25.40±3.62 | 31.64±6.49 | 24.60±4.70 | 15.48±4.67 | 14.50±4.34 | **4.78±0.67** |
| | $U$ | 24.74±6.31 | 30.91±2.62 | 24.92±4.18 | 25.04±5.48 | 15.55±8.23 | 12.45±1.86 | 15.82±4.26 | **6.80±2.81** |
| | $H$ | 34.02±5.66 | 28.92±4.11 | 23.42±4.11 | 27.34±8.17 | 19.01±4.65 | 11.42±3.72 | 12.92±4.29 | **6.02±1.32** |
| | $G$ | 25.41±4.86 | 31.49±1.86 | 23.24±2.54 | 22.31±3.45 | 31.20±4.61 | 29.72±8.79 | 13.08±4.81 | **6.10±1.33** |
| | $c_V$ | 0.19±0.01 | 0.19±0.01 | 0.19±0.00 | 0.18±0.01 | 0.18±0.02 | 0.18±0.00 | 0.19±0.03 | **0.16±0.00** |
| | $U_0^{ATOM}$ | 0.38±0.02 | 0.39±0.03 | 0.38±0.02 | 0.35±0.02 | 0.29±0.00 | 0.30±0.02 | 0.33±0.05 | **0.24±0.01** |
| | $U^{ATOM}$ | 0.40±0.04 | 0.48±0.07 | 0.40±0.04 | 0.36±0.02 | 0.30±0.01 | 0.30±0.00 | 0.33±0.06 | **0.24±0.01** |
| | $H^{ATOM}$ | 0.38±0.04 | 0.38±0.02 | 0.41±0.06 | 0.37±0.03 | 0.30±0.01 | 0.31±0.01 | 0.36±0.09 | **0.25±0.00** |
| | $G^{ATOM}$ | 0.37±0.04 | 0.34±0.02 | 0.33±0.01 | 0.31±0.02 | 0.26±0.01 | 0.27±0.01 | 0.36±0.08 | **0.22±0.02** |
| | $A$ | 0.90±0.06 | 0.97±0.12 | 1.08±0.16 | 1.01±0.10 | 64.88±29.77 | 3.82±2.05 | 1.42±0.44 | **0.75±0.11** |
| | $B$ | 0.19±0.05 | 0.21±0.04 | 0.26±0.01 | 0.26±0.02 | 0.10±0.03 | 0.11±0.01 | 0.16±0.05 | **0.08±0.01** |
| | $C$ | 0.27±0.02 | 0.27±0.01 | 0.28±0.00 | 0.28±0.01 | 0.12±0.04 | 0.10±0.02 | 0.12±0.05 | **0.05±0.01** |
| DOCK (↑) | ESR2 | 0.68±0.00 | 0.67±0.00 | 0.65±0.00 | 0.70±0.00 | OOM | 0.64±0.01 | 0.68±0.00 | **0.70±0.00** |
| | F2 | 0.89±0.00 | 0.89±0.00 | 0.89±0.00 | 0.89±0.00 | OOM | 0.87±0.01 | 0.88±0.00 | **0.89±0.00** |
| | KIT | 0.83±0.00 | 0.83±0.00 | 0.83±0.00 | **0.84±0.00** | OOM | 0.80±0.01 | 0.83±0.00 | 0.84±0.00 |
| | PARP1 | 0.92±0.00 | 0.92±0.00 | 0.92±0.00 | 0.92±0.00 | OOM | 0.91±0.00 | 0.92±0.01 | **0.93±0.00** |
| | PGR | 0.70±0.00 | 0.68±0.01 | 0.67±0.01 | 0.72±0.00 | OOM | 0.68±0.01 | 0.70±0.01 | **0.73±0.00** |
| MoleculeNet (↑) | FREESOLV | 0.97±0.00 | 0.96±0.01 | 0.97±0.01 | 0.95±0.01 | 0.93±0.00 | 0.93±0.02 | 0.86±0.03 | **0.98±0.00** |
| | LIPO | 0.81±0.01 | 0.82±0.01 | 0.82±0.01 | **0.83±0.01** | 0.61±0.04 | 0.55±0.02 | 0.79±0.00 | 0.81±0.01 |
| | ESOL | 0.94±0.01 | 0.93±0.01 | 0.93±0.00 | 0.94±0.01 | 0.91±0.02 | 0.89±0.03 | 0.91±0.00 | **0.94±0.00** |
| PCQM4MV2 (↓) | | N/A | N/A | N/A | N/A | N/A | N/A | N/A | 0.0235 |

| Dataset (↓) | GAT | GATv2 | PNA | Graphormer | TokenGT | GPS | ESA (PE) |
|---|---|---|---|---|---|---|---|
| ZINC | 0.078±0.01 | 0.079±0.00 | 0.057±0.01 | 0.036±0.00 | 0.047±0.01 | 0.024±0.01 | **0.015±0.00** |

The MAE is reported for PCQM4MV2 (it is the standard metric for this task) over a single run due to the size of the dataset. The lowest MAEs and RMSEs, the highest $R^2$ values, and table headers are highlighted in bold. OOM denotes out-of-memory errors. A complete table, including GCN and GIN, is provided in Supplementary Table 1.
Mean absolute error (MAE) on the full ZINC dataset, including an ESA model augmented with positional encodings (PE). A complete table, including GCN, GIN, and DropGIN is provided in Supplementary Table 3.

outperformed strong GNN baselines. ESA, without positional encodings, slightly underperforms compared to GraphGPS, which uses random-walk structural encodings (RWSE). This type of encoding is known to be beneficial in molecular tasks, and especially for ZINC[31,55,56]. Thus, we also evaluated an ESA + RWSE model, which increased relative performance by almost 45%. Recently, Ma et al.[36] evaluated their own method (GRIT) against 11 other baselines, including higher-order GNNs, with the best reported MAE being $0.023 \pm 0.001$. This shows that ESA can already almost match state-of-the-art models without structural encodings, and significantly improves upon this result when augmented.

**Long-range peptide tasks.** Graph learning with transformers has traditionally been evaluated on long-range graph benchmarks (LRGB)[57]. However, it was recently shown that simple GNNs outperform most attention-based methods[34]. We selected two long-range benchmarks that involve the prediction of peptide properties: PEPTIDES-STRUCT and PEPTIDES-FUNC. From the LRGB collection, these two stand out because they have the longest average shortest path ($20.89 \pm 9.79$ versus $10.74 \pm 0.51$ for the next) and the largest average diameter ($56.99 \pm 28.72$ versus $27.62 \pm 2.13$). We report ESA results against tuned GNNs and GraphGPS models from ref. 57, as well as our own optimised TokenGT model (Supplementary Table 2). Despite using only half the number of layers as other methods or less, ESA outperformed these baselines and is competitive with the best published models, with further tuning leading to state-of-the-art performance (Table 6).

**Learning on 3D atomic systems.** We adapt a subset from the Open Catalyst Project (OCP)[58,59] for evaluation, with the full steps in SI 12, including deriving edges from atom positions based on a cutoff, preprocessing specific to crystal systems, and encoding atomic distances using Gaussian basis functions. As a prototype, we compare NSA (MAE of $0.799 \pm 0.008$) against Graphormer ($0.839 \pm 0.005$), one of the best models that have been used for the Open Catalyst Project[60]. These encouraging results on a subset of OCP motivated us to further study modelling 3D atomic systems through the lens of transfer learning.

**Transfer learning on frontier orbital energies.** We follow the recipe recently outlined for drug discovery and quantum mechanics by Buterez et al.[37] and leverage a recent, refined version of the QM9 HOMO and LUMO energies[61] that provides alternative DFT calculations and new calculations at the more accurate GW level of theory. As outlined in[37], transfer learning can occur transductively or inductively. The transfer learning scenario is important and is detailed in SI 10. We used a 25K/5K/10K train/validation/test split for the high-fidelity GW data, and we trained separate low-fidelity DFT models on the entire dataset (transductive) or with the high-fidelity test set molecules removed (inductive). Since the HOMO and LUMO energies depend to a large extent on the molecular geometry, we used the 3D-aware version of ESA from the previous section and adapted all of our baselines to the 3D setup. Our results are reported in Table 2. Without transfer learning (strategy 'GW' in Table 2), ESA and PNA are almost evenly matched, which is already an improvement, since PNA was better for frontier orbital energies without 3D structures (Table 1), while graph transformers perform poorly. When using transfer learning, all methods improve significantly, but ESA outperforms all baselines for both HOMO and LUMO, in both transductive and inductive tasks.

**Mixed graph-level tasks**

Next, we examine a suite of graph-level benchmarks from various domains, including computer vision, bioinformatics, synthetic graphs, and social graphs. Our results are reported in Table 3 and

**Table 2 | A summary of the transfer learning performance on QM9 for HOMO and LUMO properties, presented as mean ± standard deviation over 5 different runs**

| Task | Strat. | DropGIN | GAT | GATv2 | PNA | Graphormer | TokenGT | GPS | ESA |
|------|--------|---------|-----|-------|-----|------------|---------|-----|-----|
| HOMO | GW | 0.162±0.00 | 0.159±0.00 | 0.157±0.00 | **0.151±0.00** | 0.179±0.01 | 0.200±0.01 | 0.162±0.00 | 0.152±0.00 |
|  | Ind. | 0.136±0.00 | 0.131±0.00 | 0.133±0.00 | 0.132±0.00 | 0.134±0.00 | 0.156±0.00 | 0.151±0.00 | **0.131±0.00** |
|  | Trans. | 0.126±0.00 | 0.123±0.00 | 0.124±0.00 | 0.121±0.00 | 0.125±0.00 | 0.137±0.00 | 0.147±0.00 | **0.119 ±0.00** |
| LUMO | GW | 0.180±0.00 | 0.181±0.00 | 0.178±0.00 | 0.174±0.00 | 0.190±0.01 | 0.204±0.01 | 0.178±0.00 | **0.174±0.00** |
|  | Ind. | 0.159±0.00 | 0.156±0.00 | 0.157±0.00 | 0.156±0.00 | 0.151±0.00 | 0.165±0.00 | 0.167±0.00 | **0.150±0.00** |
|  | Trans. | 0.157±0.00 | 0.153±0.00 | 0.153±0.00 | 0.153±0.00 | 0.147±0.00 | 0.156±0.00 | 0.169±0.00 | **0.146±0.00** |

The metric is the root mean squared error (RMSE). All models use 3D atomic coordinates and atom types as inputs, and no other node or edge features. 'Strat.' stands for strategy and specifies the type of learning: GW only (no transfer learning), inductive, or transductive. The lowest values and table headers are highlighted in bold. A complete table, including GCN and GIN, is provided in Supplementary Table 4.

**Table 3 | The table reports the Matthews correlation coefficient (MCC) for graph-level classification tasks from various domains, including molecular benchmarks from MoleculeNet (MN) and National Cancer Institute (NCI), presented as mean ± standard deviation over 5 different runs**

| | Dataset | DropGIN | GAT | GATv2 | PNA | Graphormer | TokenGT | GPS | ESA |
|---|---|---|---|---|---|---|---|---|---|
| | MalNetTiny | 0.90 ± 0.01 | 0.90 ± 0.01 | 0.90 ± 0.01 | 0.91 ± 0.01 | OOM | 0.78 ± 0.01 | 0.79 ± 0.01 | **0.93 ± 0.00** |
| Vision | MNIST | 0.97 ± 0.00 | 0.97 ± 0.00 | 0.98 ± 0.00 | 0.98 ± 0.00 | N/A | N/A | 0.98 ± 0.00 | **0.99 ± 0.00** |
| | CIFAR10 | 0.61 ± 0.01 | 0.66 ± 0.01 | 0.66 ± 0.01 | 0.69 ± 0.01 | N/A | N/A | 0.71 ± 0.01 | **0.73 ± 0.00** |
| MoleculeNet | BBBP | 0.68 ± 0.02 | 0.74 ± 0.01 | 0.73 ± 0.03 | 0.73 ± 0.03 | 0.55 ± 0.01 | 0.58 ± 0.07 | 0.70 ± 0.04 | **0.84 ± 0.01** |
| | BACE | 0.65 ± 0.03 | 0.63 ± 0.02 | 0.64 ± 0.03 | 0.64 ± 0.02 | 0.52 ± 0.02 | 0.58 ± 0.03 | 0.62 ± 0.03 | **0.72 ± 0.02** |
| | HIV | 0.46 ± 0.03 | 0.42 ± 0.06 | 0.34 ± 0.06 | 0.42 ± 0.04 | OOM | 0.46 ± 0.02 | 0.25 ± 0.21 | 0.53 ± 0.01 |
| NCI | NCI1 | 0.69 ± 0.03 | 0.70 ± 0.02 | 0.65 ± 0.03 | 0.70 ± 0.03 | 0.54 ± 0.02 | 0.53 ± 0.03 | 0.70 ± 0.03 | **0.75 ± 0.01** |
| | NCI109 | 0.68 ± 0.02 | 0.66 ± 0.01 | 0.66 ± 0.01 | 0.67 ± 0.02 | 0.50 ± 0.02 | 0.45 ± 0.03 | 0.62 ± 0.01 | **0.70 ± 0.01** |
| Bioinf. | ENZYMES | 0.58 ± 0.04 | 0.75 ± 0.02 | 0.74 ± 0.02 | 0.68 ± 0.03 | N/A | N/A | 0.73 ± 0.05 | **0.75 ± 0.01** |
| | PROTEINS | 0.46 ± 0.01 | 0.46 ± 0.04 | 0.49 ± 0.04 | 0.47 ± 0.07 | N/A | N/A | 0.44 ± 0.02 | **0.59 ± 0.02** |
| | DD | 0.54 ± 0.07 | 0.47 ± 0.05 | 0.53 ± 0.03 | 0.56 ± 0.08 | OOM | 0.46 ± 0.05 | 0.60 ± 0.05 | **0.65 ± 0.03** |
| Synthetic | SYNTHETIC | 1.00 ± 0.00 | 1.00 ± 0.00 | 1.00 ± 0.00 | 1.00 ± 0.00 | N/A | N/A | 1.00 ± 0.00 | **1.00 ± 1.00** |
| | SYNTHETICnew | 0.97 ± 0.05 | 0.76 ± 0.07 | 0.91 ± 0.07 | 1.00 ± 0.00 | N/A | N/A | 1.00 ± 0.00 | **1.00 ± 0.00** |
| | Synthie | 0.94 ± 0.02 | 0.70 ± 0.04 | 0.80 ± 0.04 | 0.88 ± 0.06 | N/A | N/A | **0.95 ± 0.02** | 0.95 ± 0.02 |
| Social | IMDB-B | 0.61 ± 0.06 | 0.69 ± 0.04 | 0.60 ± 0.05 | 0.56 ± 0.07 | 0.56 ± 0.05 | 0.61 ± 0.05 | 0.60 ± 0.05 | **0.74 ± 0.03** |
| | IMDB-M | 0.12 ± 0.10 | 0.20 ± 0.05 | 0.20 ± 0.03 | 0.03 ± 0.07 | 0.22 ± 0.03 | 0.20 ± 0.02 | 0.23 ± 0.02 | **0.25 ± 0.03** |
| | twitch_egos | 0.38 ± 0.01 | 0.37 ± 0.01 | 0.37 ± 0.01 | 0.08 ± 0.16 | 0.39 ± 0.00 | 0.39 ± 0.00 | 0.40 ± 0.00 | **0.40 ± 0.00** |
| | reddit_threads | 0.56 ± 0.01 | 0.53 ± 0.02 | 0.54 ± 0.02 | 0.11 ± 0.23 | 0.57 ± 0.00 | 0.56 ± 0.00 | **0.57 ± 0.00** | 0.57 ± 0.00 |

OOM denotes out-of-memory errors, and N/A that the model is unavailable (e.g. node/edge features are not integers, which are required for Graphormer and TokenGT). The poor performance of GraphGPS on MalNetTiny can be explained by our use of one-hot degrees as node features for datasets lacking pre-computed features, while GraphGPS originally used the more informative local degree profile[83]. The highest values and table headers are highlighted in bold. Complete tables, including GCN and GIN, are provided in Supplementary Tables 5 and 7 with MCC as the metric, and in Supplementary Tables 6 and 8 with accuracy.

Supplementary Table 8, where ESA generally outperforms all baselines. In the context of state-of-the-art models from online leaderboards, ESA already matches or competes with the best models on well-known datasets such as MNIST, MalNetTiny, and CIFAR10 without using structural or positional encodings. We also evaluated Exphormer and Polynormer on a representative selection of datasets in SI 4; however, the two baselines proved uncompetitive with ESA.

## Node-level benchmarks

Node-level tasks are an interesting challenge for our proposed approach, as the PMA module is not needed, and node-level propagation is the most natural strategy for learning node representations. To this end, we did a prototype with an edge-to-node pooling module, which would allow ESA to learn node embeddings, with good results. However, the approach does not currently scale to the millions of edges that some heterophilous graphs have (also see SI 6 and SI 7 for a discussion on extending ESA to millions of edges). For these reasons, we revert to the simpler node-set attention (NSA) formulation. Table 4 summarises our results, indicating that NSA performs well on all 11 node-level benchmarks, including homophilous (citation), heterophilous, and shortest path tasks. We have adapted Graphormer and TokenGT for node classification as this functionality was not originally available, although they require integer node and edge features, restricting their use on some datasets (denoted by N/A in Table 4). NSA achieves the highest MCC score on homophilous and heterophilous graphs, but GCN has the highest accuracy on Cora (Supplementary Table 10). Some methods, for instance GraphSAGE[62], are designed to perform particularly well under heterophily and to account for that, we include GraphSAGE[62] and Graph Transformer[39] as the two top-performing baselines from Platonov et al.[63] in our extended results in Supplementary Table 11. We also include Polynormer as another method dedicated to node classification. With a few exceptions, we outperform the baselines by a noticeable margin. Our results on CHAMELEON stand out in particular, as well as on the shortest path benchmarks, where other graph transformers are unable to learn and even PNA fails to be competitive.

## Effects of varying the layer order and type

In Table 5, we summarise the results of an ablation relative to the interleaving operator, with different numbers of layers and layer orderings. Smaller datasets perform well with 4 to 6 feature extraction layers, while larger datasets with more complex graphs, like MNIST, benefit from up to 10 layers. We have generally observed that the top configurations tend to include self-attention layers at the front, with masked attention layers in the middle and self-attention layers at the end, surrounding the PMA readout. Naive configurations such as all-masked layers or simply alternating masked and self-attention layers do not tend to be optimal for graph-level prediction tasks. This ablation experiment demonstrates the importance of vertically combining masked and self-attention layers for the performance of our model.

## Enhancing ESA for state-of-the-art performance

So far, we have focused on establishing that ESA is a strong foundation for graph learning and have not leveraged positional and structural encodings or other helper mechanisms (with the exception of ZINC in Table 1). These models were already outperforming or matching the state-of-the-art on several datasets. To explore the potential and compatibility of ESA with additional transformations and components, we further tune hyperparameters such as the optimiser and explore the utility of structural encodings[56] and/or problem-specific modifications of input graphs (i.e. explicit incorporation of ring membership as node features for ZINC[64]). For clarity, we refer to this extension as ESA+ and report our results in Table 6. The results on all six datasets are state-of-the-art and top the online leaderboards at the time of writing (March 2025).

**Table 4 | The table reports the Matthews correlation coefficient (MCC) for 11 node-level classification tasks, presented as mean ± standard deviation over 5 different runs**

| | Dataset (↑) | GCN | GAT | GATv2 | PNA | Graphormer | TokenGT | GPS | NSA |
|---|---|---|---|---|---|---|---|---|---|
| | PPI | 0.98 ± 0.00 | 0.99 ± 0.00 | 0.98 ± 0.01 | 0.99 ± 0.00 | N/A | N/A | N/A | **0.99 ± 0.00** |
| Citation | CiteSeer | 0.61 ± 0.01 | 0.59 ± 0.03 | 0.61 ± 0.01 | 0.51 ± 0.03 | OOM | 0.38 ± 0.02 | 0.54 ± 0.01 | **0.63 ± 0.00** |
| | Cora | 0.77 ± 0.01 | 0.75 ± 0.01 | 0.73 ± 0.01 | 0.64 ± 0.03 | OOM | 0.37 ± 0.18 | 0.64 ± 0.04 | **0.77 ± 0.00** |
| Heterophily | ROMAN EMPIRE | 0.47 ± 0.00 | 0.74 ± 0.01 | 0.76 ± 0.00 | 0.86 ± 0.00 | N/A | N/A | 0.84 ± 0.01 | **0.87 ± 0.00** |
| | AMAZON RATINGS | 0.18 ± 0.00 | 0.26 ± 0.01 | 0.25 ± 0.01 | 0.21 ± 0.02 | N/A | N/A | 0.11 ± 0.14 | **0.34 ± 0.01** |
| | MINESWEEPER | 0.30 ± 0.00 | 0.48 ± 0.02 | 0.51 ± 0.01 | 0.62 ± 0.04 | N/A | N/A | 0.56 ± 0.01 | **0.69 ± 0.00** |
| | TOLOKERS | 0.30 ± 0.01 | 0.38 ± 0.01 | 0.39 ± 0.00 | 0.35 ± 0.05 | N/A | N/A | 0.35 ± 0.02 | **0.43 ± 0.00** |
| | SQUIRREL | 0.20 ± 0.01 | 0.24 ± 0.01 | 0.24 ± 0.02 | 0.22 ± 0.01 | 0.10 ± 0.09 | 0.18 ± 0.02 | 0.23 ± 0.02 | **0.29 ± 0.01** |
| | CHAMELEON | 0.32 ± 0.01 | 0.24 ± 0.06 | 0.28 ± 0.03 | 0.27 ± 0.03 | 0.25 ± 0.04 | 0.26 ± 0.04 | 0.30 ± 0.08 | **0.39 ± 0.02** |
| Shortest path | ER (15K) | 0.22 ± 0.02 | 0.32 ± 0.00 | 0.32 ± 0.00 | 0.54 ± 0.09 | OOM | 0.06 ± 0.00 | 0.18 ± 0.04 | **0.92 ± 0.01** |
| | ER (30K) | 0.09 ± 0.03 | 0.10 ± 0.06 | 0.10 ± 0.06 | 0.42 ± 0.05 | OOM | OOM | OOM | **0.87 ± 0.01** |

The number of nodes for the shortest path benchmarks is given in parentheses (based on randomly-generated infected Erdős-Rényi (ER) graphs; details are provided in SI 11). For SQUIRREL and CHAMELEON, we used the filtered datasets introduced by Platonov et al.[63] to fix existing data leaks. The highest values and table headers are highlighted in bold. Additional heterophily results are provided in Supplementary Table 11. A complete table, including GIN and DropGIN, is provided in Supplementary Table 9 with MCC as the performance metric, and in Supplementary Table 10 with accuracy.

## Time and memory scaling

The technical aspects of ESA are discussed in Methods, where we also cover deep learning library limitations and possible optimisations (also see SI 6 and SI 7). Although efficient attention implementations have linear memory complexity, storing the edge adjacency matrices in a dense format is currently a bottleneck imposed by available frameworks. Here, we have empirically evaluated the time and memory scaling of ESA against all 9 baselines. For a fair evaluation, we implemented Flash attention for TokenGT as it was not originally supported. GraphGPS natively supports Flash attention, while Graphormer requires specific modifications to the attention matrix that are not currently supported.

We report the training time for a single epoch and the maximum allocated memory during training for QM9 and MNIST in Fig. 2A to D, and DOCKSTRING in Supplementary Fig. 1. In terms of training time, GCN and GIN are consistently the fastest due to their simplicity and low number of parameters (also see Fig. 2G). ESA is usually the next fastest, followed by other graph transformers. The strong GNN baselines, particularly PNA, and to an extent GATv2 and GAT, are among the slowest methods. In terms of memory, GCN and GIN are again the most efficient, followed by GraphGPS and TokenGT. ESA is only slightly behind, with PNA, DropGIN, GATv2, and GAT all being more memory-intensive, particularly DropGIN.

We also order all methods according to their achieved performance and illustrate their rank relative to the total training time and the maximum memory allocated during training (Fig. 2E, F). ESA occupies the top left corner in the time plot, confirming its efficiency. The results are more spread out regarding memory; however, the performance is still remarkable considering that ESA works over edges, whose number rapidly increases relative to the number of nodes that dictates the scaling of all other methods.

Finally, we investigate and report the number of parameters for all methods and their configurations (Fig. 2G). Apart from GCN, GIN, and DropGIN, ESA has the lowest number of parameters. GAT and GATv2 have rapidly increasing parameter counts, likely due to the concatenation that occurs between attention heads, while PNA also increases significantly quicker than ESA. Lastly, we demonstrate and discuss one of the bottlenecks in Graphormer for the dataset HIV (Fig. 2H). This lack of efficiency is likely caused by the edge features computed by Graphormer, which depend quadratically on the number of nodes in the graph and on the maximum shortest path in the graph.

## Explainability

Explainability is one aspect where attention models excel compared to other architectures, and since ESA is fundamentally edge-based, analysing attention scores might lead to different insights compared to traditional models. Here, we investigate two ways of interpreting the learnt attention weights/scores of ESA and visualise the results based on the QM9 model from Table 1, using the following architecture: `MMMMMSPS`. This choice is motivated by the availability of several quantum properties with significantly different physical characteristics, and in particular (1) HOMO (highest occupied molecular orbital) energy, an intensive and localised property, and (2) u0 (internal energy at 0K), an extensive and global property. As the HOMO can be highly localised in a region of the molecule, while the internal energy should be evenly distributed across all atoms, we expect this dynamic to be reflected in the weights learnt by the model. More specifically, in the case of HOMO we hypothesise that only a few attention scores would dominate for each molecule, while for u0 the scores should be dispersed relatively uniformly over all the atoms.

To quantify this notion, we leveraged the Gini coefficient, a measure of inequality that ranges from 0 to 1. A value of 0 indicates equality among all values, while a value of 1 indicates maximal inequality, where a single value dominates and the rest are 0. Our results are illustrated in Fig. 3 for all 7 encoder layers and show that while the attention scores start from an even spread for both HOMO and u0, in the case of HOMO we notice progressively more peaked and dominant scores until the last layer, while for u0 the opposite is true, with the latter layers exhibiting the most dispersed attention.

For our second analysis, we investigated whether the bonds corresponding to the top attention scores of the HOMO model contribute to the ground truth molecular orbitals calculated from the 3D conformations available in the QM9 dataset. Our protocol and visualisations are covered in SI 16, showing that the model indeed learns physically relevant relationships for a variety of molecules.

## Discussion

We presented an end-to-end attention-based approach for learning on graphs and demonstrated its effectiveness relative to tuned graph neural networks and recently proposed graph transformer architectures. The approach is free of expensive pre-processing steps and

**Table 5 | Example of multiple ESA configurations and their impact on performance via three different kinds of benchmarks: a graph-level molecular task (BBBP), a graph-level vision task (MNIST), and a node-level heterophilous task (CHAMELEON)**

| Dataset | Model | MCC (↑) |
|---|---|---|
| BBBP | M M M M S P S | 0.845 |
| | M M M S P S S | 0.835 |
| | S S S M M P S | 0.812 |
| | M M M M M P | 0.782 |
| | S M S M S P | 0.768 |
| MNIST | S S M M M M M M P S | 0.986 |
| | S M S M S M S M P | 0.983 |
| | S S S M M M S S S P | 0.982 |
| | M S M S M S M S P | 0.980 |
| | M M M M M M M M P | 0.980 |
| CHAMELEON | M S M S M S | 0.422 |
| | S S M M S S | 0.384 |
| | S M S M S M | 0.378 |
| | M M M M M | 0.359 |
| | S S M M M M | 0.351 |

In model configurations, M denotes a MAB, S a SAB, and P is the PMA module. We illustrate the top-performing configurations and their variations, as well as various patterns such as only masked layers or masked/self-attention alterations. The performance metric is the Matthews correlation coefficient (MCC). Table headers are highlighted in bold.

**Table 6 | The table reports the performance of tuned ESA+ models on a selection of six datasets, with different evaluation metrics to facilitate comparisons across published papers and leaderboards**

| Model | ZINC | ZINC-FULL | PEPTIDES-STRUCT | PEPTIDES-FUNC | MNIST | MalNetTiny |
|---|---|---|---|---|---|---|
| Metric | MAE (↓) | MAE (↓) | MAE (↓) | AP (↑) | Accuracy (↑) | Accuracy (↑) |
| ESA+ | $0.052 \pm 0.001$ | $0.011 \pm 0.000$ | $0.239 \pm 0.000$ | $0.736 \pm 0.004$ | $98.917 \pm 0.020$ | $94.800 \pm 0.424$ |

The results for the full version of ZINC (with 250,000 compounds) and a commonly used subset of ZINC with 12,000 compounds are reported separately. Molecular models leverage random-walk structural encodings (RWSE). Table headers are presented in bold.

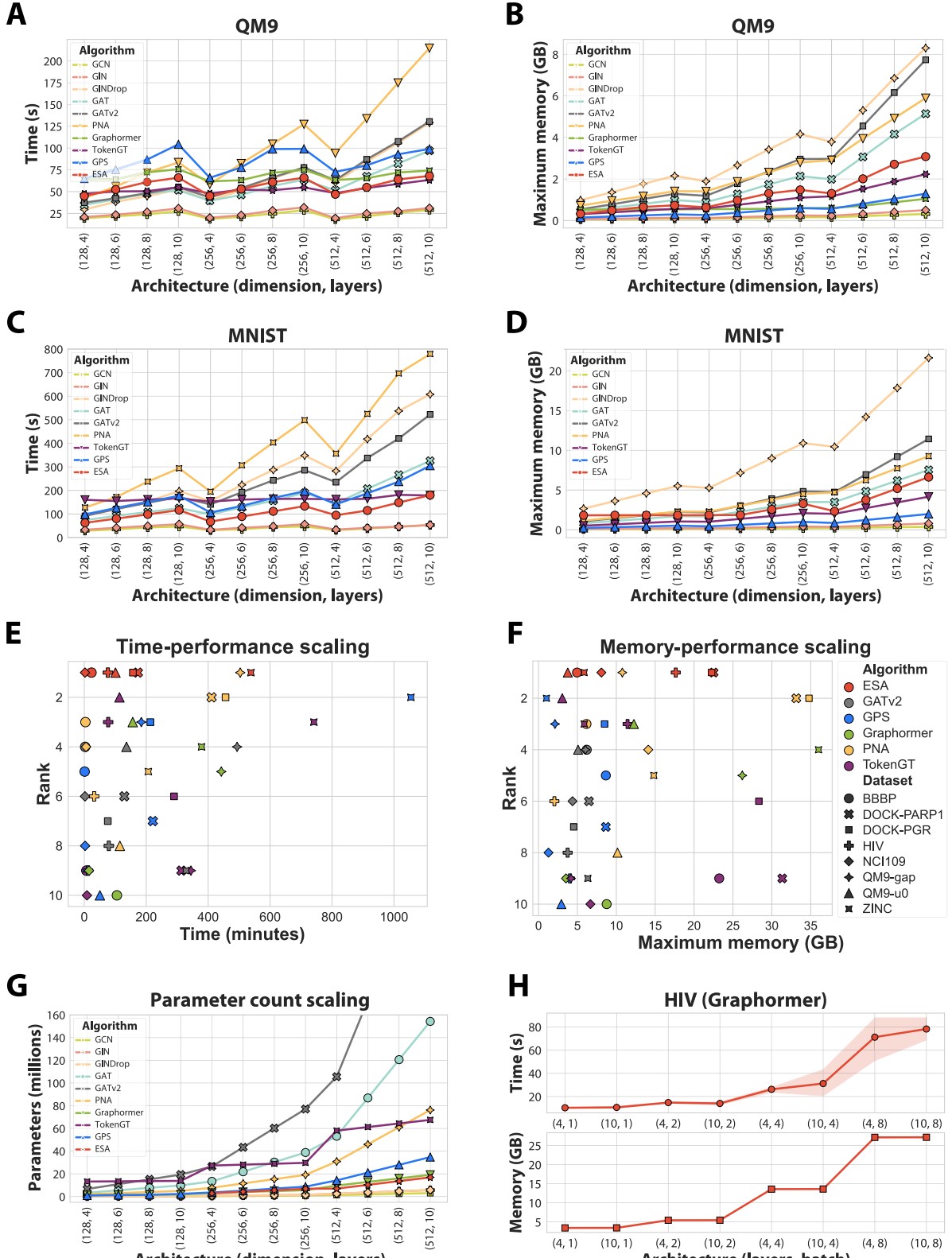

**Fig. 2 | Time, memory, and parameter statistics for all evaluated models. A–D** The elapsed time for training all evaluated models for a single epoch and for different datasets (in seconds), and the maximum allocated memory during this training epoch (GB), while varying hidden dimensions and the number of layers. QM9 has around 130K small molecules (a maximum of 29 nodes and 56 edges), DOCKSTRING (Supplementary Fig. 1) has around 260K graphs that are around 6 times larger, and finally MNIST has 70K graphs, which are around 11-12 times larger than QM9. We use dummy integer features for TokenGT when benchmarking

MNIST. Graphormer runs out of memory for DOCKSTRING and MNIST. **E**, **F** All methods illustrated according to their achieved performance (rank) versus the total time spent training (**E**) and the maximum allocated memory (**F**). **G** The number of parameters (in millions) for all the methods and different configurations in terms of the hidden dimension and the number of layers. **H** The time and memory bottleneck in Graphormer. Even training on a single graph from HIV takes over 10 seconds and slightly under 5 GB. Extrapolating these numbers for a batch size of 8 to the entire dataset results in a training time of 5 days for a single epoch.

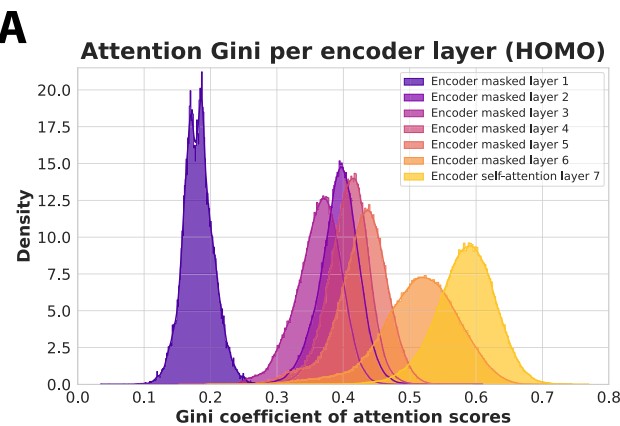

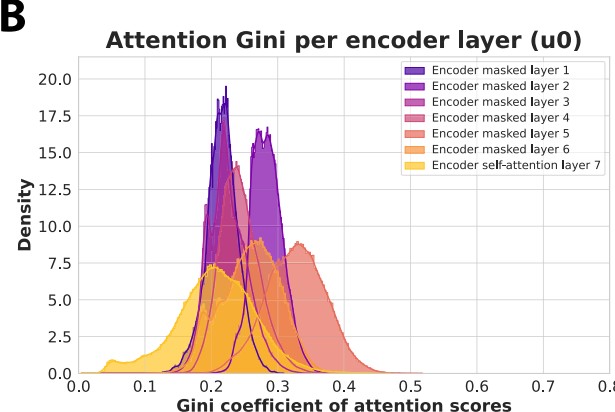

**Fig. 3 | Per-layer visualisation of the learnt attention scores on quantum properties from the QM9 dataset. A**, **B** Histograms of the Gini coefficients for attention scores on two quantum properties: HOMO (highest occupied molecular orbital energy) in **A** and u0 (internal energy at 0K) in **B**. The attention scores for each layer are first averaged over the number of heads.

numerous other additional components designed to improve the inductive bias or expressive power (e.g. shortest paths, centrality, spatial and structural encodings, virtual nodes as readouts, expander graphs, transformations via interaction graphs, etc.). This shows huge potential and plenty of room for further fine-tuning and task-specific improvements. The interleaving operator inherent to ESA allows for vertical combination of masked and self-attention modules for learning effective token (i.e. edge or node) representations, leveraging relational information specified by input graphs while at the same time allowing to expand on this prior structure via self-attention.

ESA utilises line graphs[65] where the edges from an input graph are transformed into nodes in the line graph, establishing connections if they share a common node in the original graph. This transformation preserves isomorphisms, meaning that if two graphs are isomorphic, their line graphs are as well; however, the reverse is not necessarily true. Notably, some non-isomorphic graphs that are indistinguishable by the 1-WL test become distinguishable through their line graph embeddings (see SI 17 and Supplementary Fig. 4), possibly highlighting richer information conveyed by edge tokens as opposed to node tokens in some tasks. Line graphs naturally emphasise important motifs[65–67] like triangles, cycles, and cliques, which are critical for tasks such as predicting molecular properties. Furthermore, line graphs tend to balance vertex degree variance, maintaining a more uniform distribution, which enhances the regularity of message passing. This edge-focused communication strategy aligns with previous explorations in GNNs, such as in Chemprop[68], which hypothesised improved accuracy resulting from handling information propagation more effectively than node-centric message passing.

Our comprehensive evaluation shows that the proposed approach consistently outperforms strong message passing baselines and recently proposed transformer-based approaches for learning on graphs. The takeaway from our extensive study is that the proposed approach is well-suited to be a simple yet extremely effective starting point for learning on graphs. Indeed, preliminary results on enhancing ESA with positional encodings indicate state-of-the-art results on several datasets. Moreover, the approach has favourable computational complexity and scales better than strong GNNs and some graph transformers. ESA also does well in transfer learning settings, possibly paving the way for more research on foundational models for structured data and drug discovery in particular, where the problem arises frequently in property prediction for expensive high-fidelity experiments.

The widespread approach of storing attention masks as dense tensors is likely caused by the limited interest in structured masking, since the main driving force behind efficient attention

implementations is scaling up large language models (LLMs), where the only masking requirements are simple causal masks or similar pre-defined patterns. In principle, one only needs to load and apply the masking information per chunk, and in practice, this manipulation would lead only to a small overhead while avoiding hundreds of millions of computations. Such an implementation is not trivial, as it requires knowledge of GPU architectures and low-level programming abstractions like Triton[69] and CUDA[70]. As such, it is not within the scope of our current work. The very recent Flex Attention project[71] is partially motivated by our initiative and aims to extend and streamline efficient attention operations, including sparse patterns. However, such prototypes are still not fully functional. Nevertheless, even our current implementation covers a wide diversity of application domains, such as computational chemistry (small molecules and peptides), computer vision, bioinformatics, social networks, and many others.

## Methods
In this section, we provide a high-level overview of our architecture and describe the masked attention module that allows information to propagate across edges as primitives. An alternative implementation for the more traditional node-based propagation is also presented. The section starts with a formal definition of the masked self-attention mechanism and then proceeds to describe our end-to-end attention-based approach to learning on graphs, involving the encoder and the pooling components (illustrated in Fig. 4).

### Masked attention modules
We first introduce the basic notation and then give a formal description of the masked attention mechanism with a focus on the connectivity pattern specific to graphs. Following this, we provide algorithms for an efficient implementation of masking operators using native tensor operations.

As in most graph learning settings, a graph is a tuple $\mathcal{G} = (\mathcal{N}, \mathcal{E})$ where $\mathcal{N}$ represents the set of nodes (vertices), $\mathcal{E} \subseteq \mathcal{N} \times \mathcal{N}$ is the set of edges, and $N_n = |\mathcal{N}|$, $N_e = |\mathcal{E}|$. The nodes are associated with feature vectors $\mathbf{n}_i$ of dimension $d_n$ for all nodes $i \in \mathcal{N}$, and $d_e$-dimensional edge features $\mathbf{e}_{ij}$ for all edges $(i,j) \in \mathcal{E}$. The node features are collected as rows in a matrix $\mathbf{N} \in \mathbb{R}^{N_n \times d_n}$, and similarly for edge features in $\mathbf{E} \in \mathbb{R}^{N_e \times d_e}$. The graph connectivity can be specified as an adjacency matrix $\mathbf{A}$, where $\mathbf{A}_{ij} = 1$ if $(i,j) \in \mathcal{E}$ and $\mathbf{A}_{ij} = 0$ otherwise. The edge list (edge index) representation is equivalent but more common in practice.

The main building block in ESA is masked scaled dot product attention. Panel A in Fig. 4 illustrates this attention mechanism

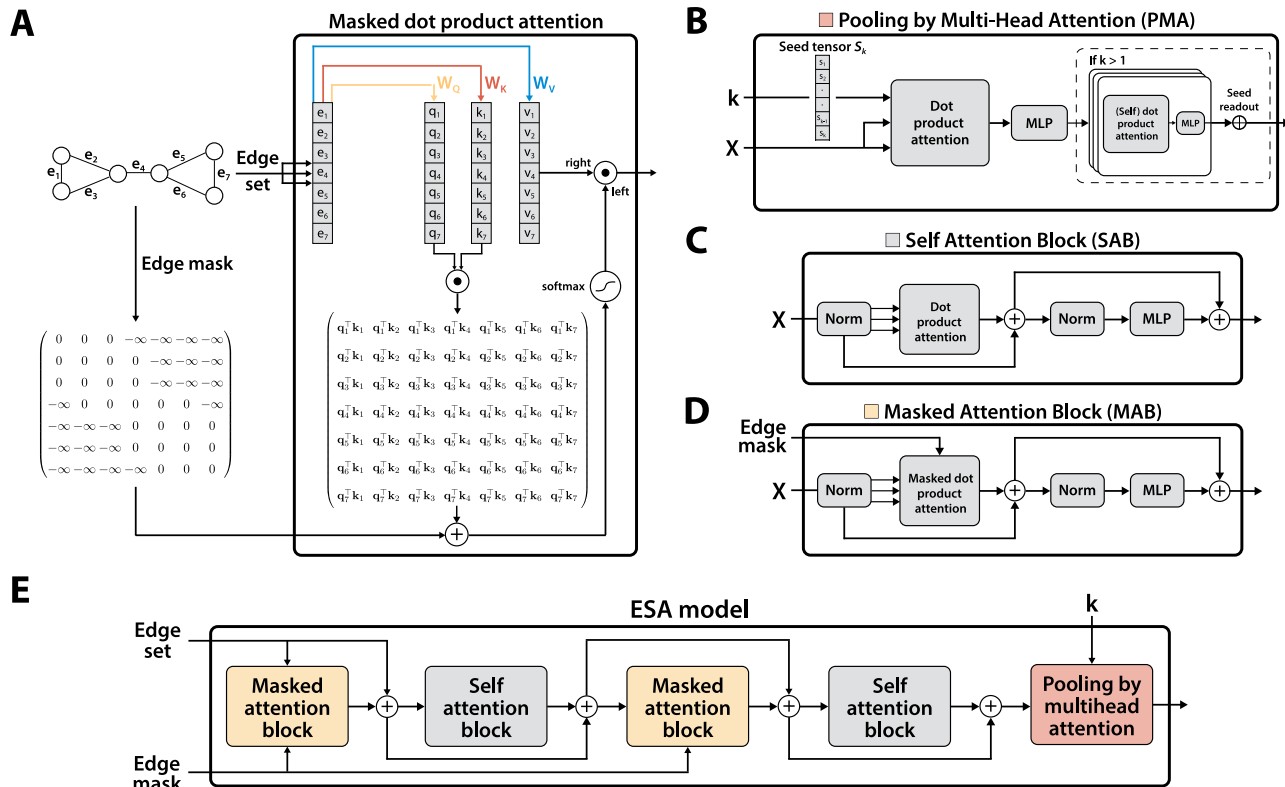

**Fig. 4 | A high-level overview of Edge Set Attention and its main building blocks. A** An illustration of masked scaled dot product attention with edge inputs. Edge features derived from the edge set are processed by query, key, and value projections as in standard attention. An additive edge mask, derived as in Algorithm 1, is applied prior to the softmax operator. A mask value of 0 leaves the attention score unchanged, while a large negative value completely discards that attention score. **B** The Pooling by Multi-Head Attention Block (PMA). A learnable set of seed tensors $\mathbf{S}_k$ is randomly initialised and used as the query for a cross-attention operation with the inputs. The result is further processed by Self-Attention Blocks. **C** The Self Attention Block (SAB), illustrated here with a pre-layer-normalisation architecture. The input is normalised, processed by self-attention, then further normalised and processed by an MLP. The block uses residual connections. **D** Illustration of the Masked (Self) Attention Block (MAB). The only difference compared to SAB is the edge mask, which follows the computation outlined in panel (**A**). **E** An instantiation of the ESA model. Edge inputs are processed by various different attention blocks, here in the order `MSMSP`, corresponding to masked, self, masked, self, and pooling attention layers.

schematically. More formally, this attention mechanism is given by

$$\mathrm{SDPA}\,(\mathbf{Q},\mathbf{K},\mathbf{V},\mathbf{M}) = \mathrm{softmax}\left(\frac{\mathbf{Q}\mathbf{K}^{\top}}{\sqrt{d_k}} + \mathbf{M}\right)\mathbf{V} \qquad (1)$$

where $\mathbf{Q}$, $\mathbf{K}$, and $\mathbf{V}$ are projections of the edge representations to queries, keys, and values, respectively. This is a minor extension of the standard scaled dot product attention via the additive mask $\mathbf{M}$ that can censor any of the pairwise attention scores, and $d_k$ is the key dimension. The generalisation to masked multi-head attention follows the same steps as in the original transformer[26]. In the following sections, we refer to this function as MultiHead($\mathbf{Q},\mathbf{K},\mathbf{V},\mathbf{M}$).

Masked self-attention for graphs can be seen as graph-structured attention, and an instance of a generalised attention mechanism with custom attention patterns[32]. More specifically, in ESA, the attention pattern is given by an edge adjacency matrix rather than allowing for interactions between all set items. Crucially, the edge adjacency matrix can be efficiently computed both for a single graph and for batched graphs using exclusively tensor operations. The case for a single graph is covered on the left-hand side of Algorithm 1 through the `edge_adjacency` function. The first 3 lines of the function correspond to getting the number of edges, then separating the source and target nodes from the edge adjacency list (also called edge index), which is equivalent to the standard adjacency matrix of a graph. The source and target node tensors each have a dimension equal to the number of edges ($N_e$). Lines 7-9 add an additional dimension and efficiently repeat (`expand`) the existing tensors to shape ($N_e$, $N_e$) without allocating new memory. Using the transpose, line 11 checks if the source nodes of any

two edges are the same, and the same for target nodes on line 12. On line 13, cross connections are checked, where the source node of an edge is the target node of another, and vice versa. The operations of lines 11-14 result in boolean matrices of shape ($N_e$, $N_e$), which are summed for the final returned edge adjacency matrix. The right-hand side panel of Algorithm 1 shows the case where the input graph represents a batch of smaller graphs. This requires an additional batch mapping tensor that maps each node in the batched graph to its original graph, and carefully manipulating the indices to create the final edge adjacency mask of shape ($B$, $L$, $L$), where $L$ is the maximum number of edges in the batch.

Since in ESA attention is computed over edges, we chose to separate source and target node features for each edge, similarly to lines 4-5 of Algorithm 1, and concatenate them to the edge features:

$$\mathbf{x}_{ij} = \mathbf{n}_i \parallel \mathbf{n}_j \parallel \mathbf{e}_{ij} \qquad (2)$$

for each edge $e_{ij}$. The resulting features $\mathbf{x}_{ij}$ are collected in a matrix $\mathbf{X} \in \mathbb{R}^{N_e \times (2d_n + d_e)}$.

Having defined the mask generation process and the masked multi-head attention function, we next define the modular blocks of ESA, starting with the Masked Self Attention Block (MAB):

$$\mathrm{MAB}\,(\mathbf{X},\mathbf{M}) = \mathbf{H} + \mathrm{MLP}(\mathrm{LayerNorm}(\mathbf{H})) \qquad (3)$$

$$\mathbf{H} = \overline{\mathbf{X}} + \mathrm{MultiHead}\,(\overline{\mathbf{X}},\overline{\mathbf{X}},\overline{\mathbf{X}},\mathbf{M}) \qquad (4)$$

$$\overline{\mathbf{X}} = \mathrm{LayerNorm}\,(\mathbf{X}) \qquad (5)$$

where the mask $\mathbf{M}$ is computed as in Algorithm 1 and has shape $B \times L \times L$, with $B$ the batch size, and MLP is a multi-layer perceptron. A self-attention block (SAB) can be formally defined as:

$$\text{SAB}(\mathbf{X}) = \text{MAB}(\mathbf{X}, \mathbf{0}) \quad (6)$$

In principle, the masks can be arbitrary and a promising avenue of research could be in designing new mask types. For certain tasks, such as node classification, we also defined an alternative version of ESA for nodes (NSA). The only major difference is the mask generation step. The single graph case is trivial, as all the masking information is available in the edge index, so we provide the general batched case in Algorithm 2.

## ESA architecture

Our architecture consists of two components: (*i*) an encoder that interleaves masked and self-attention blocks to learn an effective representation of edges, and (*ii*) a pooling block based on multi-head attention, inspired by the decoder component from the set transformer architecture[72]. The latter comes naturally when one considers graphs as sets of edges (or nodes), as is done here. ESA is a purely attention-based architecture that leverages the scaled dot product mechanism proposed by Vaswani et al.[26] through masked and standard self-attention blocks (MABs and SABs) and a pooling by multi-head attention (PMA) block.

The encoder, consisting of arbitrarily interleaved MAB and SAB blocks, is given by:

$$\text{Encoder}(\mathbf{X}, \mathbf{M}) = \text{AB} \circ \text{AB} \circ \cdots \circ \text{AB}(\mathbf{X}, \mathbf{M}), \quad \text{where} \quad \text{AB} \in \{\text{MAB}, \text{SAB}\} \quad (7)$$

Here, AB refers to an attention block that can be instantiated as an MAB or SAB.

The pooling module that is responsible for aggregating the processed edge representations into a graph-level representation is formally defined by:

$$\text{PMA}_{k,p}(\mathbf{Z}) = \text{SAB}^p(\overline{\mathbf{S}} + \text{MLP}(\overline{\mathbf{S}})) \quad (8)$$

$$\overline{\mathbf{S}} = \text{LayerNorm}(\text{MultiHead}(\mathbf{S}_k, \mathbf{Z}, \mathbf{Z}, \mathbf{0})) \quad (9)$$

where $\mathbf{S}_k$ is a tensor of $k$ learnable seed vectors that are randomly initialised and $\text{SAB}^p(\cdot)$ is the application of $p$ SABs. Technically, it suffices to set $k = 1$ to output a single representation for the entire graph. However, we have empirically found it beneficial to set it to a small value, such as $k = 32$. Moreover, this change allows self-attention (SABs) to further process the $k$ resulting representations, which can be simply summed or averaged due to the small $k$. Contrary to classical readouts that aggregate directly over set items (i.e. nodes), pooling by multi-head attention performs the final aggregation over the embeddings of learnt seed vectors $S_k$. Whereas tasks involving node-level predictions require only the Encoder component, predictive tasks involving graph-level representations require all the modules, both the encoder and pooling by multi-head attention. The architecture in the latter setting is formally given by

$$\mathbf{Z}_{\text{out}} = \text{PMA}_{k,p}(\text{Encoder}(\mathbf{X}, \mathbf{M}) + \mathbf{X}) \quad (10)$$

As optimal configurations are task-specific, we do not explicitly fix all the architectural details. For example, it is possible to select between layer and batch normalisation, a pre-LN or post-LN architecture[73], or standard and gated MLPs, along with GLU variants, e.g. SwiGLU[74].

## Time and memory scaling

ESA is enabled by recent advances in efficient and exact attention, such as memory-efficient attention and Flash attention[75–78]. Memory-efficient implementations with arbitrary masking capabilities exist in both PyTorch and xFormers[79,80]. Efficient attention avoids materialising the $N \times N$ square attention matrix, instead working over smaller chunks and saving partial aggregates that can recover the actual attention scores. Thanks to chunking and several optimisations, exact attention can be computed with linear memory requirements. Furthermore, although in theory the time complexity is still given by a quadratic number of computations, in practice GPU optimisations such as chunking, fused kernels, reduced memory bottlenecks (better bandwidth utilisation), and others can lead to forward passes that are up to $\times 25$ times faster, with comparable uplifts for the backward pass[77,78]. The complexity of ESA directly corresponds to that of the underlying attention implementation, since that is the main learning mechanism, augmented with relatively cheap MLPs and normalisation layers. A precise complexity analysis would have to include specific GPU architecture details that change from one GPU type to another, as well as a per-case discussion based on the specific chunk size, sequence length (here, number of edges), total hidden dimension, and number of heads. This is because the kernels are available only for certain combinations of sequence lengths, dimensions, and heads, and they differ in performance.

All attention operations in ESA, except the cross-attention in PMA, are based on self-attention, and our method directly benefits from all of these advances. Moreover, our cross-attention is performed between the full set of size $L$ and a small set of $k \leq 32$ learnable seeds, such that the complexity of the operation is $O(kL)$ even for standard attention. Indeed, currently, the main bottleneck is not the attention computation itself, but storing the dense edge adjacency masks. Unfortunately, all the available libraries that support masking and efficient attention kernels require a dense matrix mask, which demands memory quadratic in the number of edges. An evident, but not yet available, optimisation could amount to storing the masks in a sparse tensor format. To visualise the benefits of this optimisation, we estimate the cost of storing the sparse edge adjacency information in addition to end-to-end training a model with a full self-attention layer (no masking). The performance of this hypothetical model is visualised in SI 6 and shows that ESA could scale to graphs of 2.6 million edges on a single GPU with 40 GB of memory, even without masking hundreds of millions of unnecessary calculations. Despite some of the current limitations, we have successfully trained ESA models with batch sizes of 8 graphs with approximately 30,000 edges each (e.g. DD in Table 3), roughly equivalent to a single graph with 240,000 edges. Further limitations and possible optimisations are discussed in SI 7.

## Data availability

All the datasets used throughout the paper are publicly available through different hosting services, as indicated in the main text and supplementary materials. The majority of datasets are accessed via PyTorch Geometric. Other sources include DOCKSTRING[81], heterophily datasets, accurate GW frontier orbital energies of 134 kilo molecules of the QM9 dataset[82], Open Catalyst Project. All instances of synthetic data are reproducibly generated by the code available in our GitHub repository; processed datasets (e.g. infected Erdős-Rényi graphs) are also available in our repository.

## Code availability

The source code that enables all experiments to be reproduced, as well as code documentation, examples, and set-up instructions are available in our GitHub repository: https://github.com/davidbuterez/edge-set-attention.

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

## Acknowledgements
We are thankful for the PhD sponsorship awarded to D.B. and access to the Scientific Computing Platform within AstraZeneca.

## Author contributions
D.B., J.P.J., D.O., and P.L. were involved in conceptualising and reviewing the manuscript. D.B. designed and carried out experiments and produced the figures and tables. J.P.J., D.O., and P.L. jointly supervised the work.

## Competing interests
David Buterez has been supported by a fully funded PhD grant from AstraZeneca. J.P.J. and D.O. are employees and shareholders at AstraZeneca. All other authors declare no competing interests.
