## [Transparent Peer Review file · Nature Communications]

An end-to-end attention-based approach for learning on graphs

Corresponding Author: Mr David Buterez

Version 0:

Reviewer comments:

Reviewer #1

(Remarks to the Author)

The authors can find my comments in the attached PDF.

(Remarks on code availability)

Code was not available, therefore I was not able to check the validity of the experiments.

Reviewer #3

(Remarks to the Author)

The manuscript presents an end-to-end attention-based architecture for learning on graphs, addressing limitations in existing Graph Neural Networks (GNNs) and transformer-based models. The proposed Edge-Set Attention (ESA) model treats graphs as sets of edges rather than nodes, interleaving masked and self-attention layers in its encoder, followed by an attention-based pooling mechanism. The study benchmarks ESA on over 70 datasets across various domains, demonstrating superior performance compared to tuned GNNs and recent transformer-based methods.

Overall, the paper is interesting and well-presented. However, several issues should be addressed before the manuscript is ready for publication. See below:

1. The decision to treat graphs as edge sets rather than node-centric structures is interesting but somewhat unconventional. Could the authors further justify why an edge-centric representation is fundamentally superior? Are there theoretical guarantees that ESA can universally capture graph structure better than node-based GNNs and traditional graph transformer methods?
2. Although ESA utilizes efficient attention computation, it still requires memory quadratic in the number of edges to maintain dense adjacency masks. How does ESA handle extremely large graphs (e.g., graphs with millions of edges)? The paper states that ESA can handle up to 30,000 edges, but is it still scalable beyond this size? Could sparse attention mechanisms (e.g., Performer [1], Longformer-style [2] attention) be further optimized for ESA? Besides, it is recommended to provide a detailed complexity analysis.
3. Some state-of-the-art models are missing from comparisons. While ESA is compared against several GNNs and transformers (Graphormer, TokenGT, GPS, etc.), how does ESA compare to Expormer [3], Polynormer [4], and some pre-training methods for GNNs like GraphMAE [5]?
4. Explainability. Given the increasing focus on explainable AI, how interpretable are ESA's learned representations? Could attention maps be visualized to provide insight into the model's decision-making process?
5. ESA focuses on edge-centric representations. How does ESA perform on graphs where edges are sparse or noisy?
6. How sensitive is ESA to hyperparameter choices (e.g., number of layers, embedding dimension, attention heads)?

Minor comments:

1. Many acronyms such as ESA, MAB, SAB, PMA, NSA, are introduced quickly. It is recommended to provide clear definitions when they first appear.

[1] Choromanski, Krzysztof Marcin, et al. "Rethinking Attention with Performers." International Conference on Learning

Representations. 2020.

[2] Beltagy, Iz, Matthew E. Peters, and Arman Cohan. "Longformer: The long-document transformer." arXiv preprint arXiv:2004.05150. 2020.

[3] Shirzad, Hamed, et al. "Exphormer: Sparse transformers for graphs." International Conference on Machine Learning. PMLR, 2023.

[4] Deng, Chenhui, Zichao Yue, and Zhiru Zhang. "Polynormer: Polynomial-Expressive Graph Transformer in Linear Time." The Twelfth International Conference on Learning Representations. 2024.

[5] Hou, Zhenyu, et al. "Graphmae: Self-supervised masked graph autoencoders." Proceedings of the 28th ACM SIGKDD Conference on Knowledge Discovery and Data Mining. 2022.

(Remarks on code availability)

Version 1:

Reviewer comments:

Reviewer #1

(Remarks to the Author)

The authors thoroughly addressed all the issues I have raised in the first revision, therefore I suggest this manuscript for acceptance in this journal.

(Remarks on code availability)

I want to apologize with the authors for not having seen the available code in the first place. Code is well written and well documented, and results are reproducible.

Reviewer #3

(Remarks to the Author)

1. The authors have addressed all of my concerns.

(Remarks on code availability)

ESA: An end-to-end attention-based approach for learning on graphs

Point-by-point response

We would like to thank the Editor and the Reviewers for their time, insights, and constructive feedback. We think our manuscript has been received well and we are dedicated to improving it and ESA further. To this end, we have addressed all the remarks, including additional experiments where relevant. These have also been incorporated in our revised manuscript.

Code availability

It was noted as part of the review process that the source code for ESA is not available. We did, in fact, include a source code archive as part of the original submission to the journal, including not only the source itself but also extensive documentation, examples, and set-up instructions. We apologise if the availability of the code was not clear in our original submission, and the full code will also be available in the revised submission and shortly on GitHub.

Reviewer 1

Comment 1 (typos)

There are some typos in the manuscript (e.g., end of page 2, "Attention coefficients there are defined..."); I would suggest to revise it from a grammar and syntax point of view;

Response

We have corrected the highlighted typo and reviewed the entire manuscript to improve clarity and fix errors.

Comment 2 (oversmoothing and oversquashing)

In the "Introduction" section the authors talk about the well-known issues of oversmoothing and oversquashing; nevertheless, in my opinion very important references in literature concerning these concepts are not addressed, e.g. [1, 2, 3]; in general, there is a lack of comparison with these energy-based graph learning models;

Response

We have rectified this omission in our revised manuscript and included a discussion of the suggested papers in the relevant place in the Introduction. While being notable as alternatives to conventional graph neural networks and the well-known message passing paradigm, these contributions aim to characterize oversmoothing and oversquashing, which are often studied through homophilous and heterophilous node classification tasks. We would like to note that our work is primarily focused on graph-level predictive tasks, typically encountered in applications such as computational chemistry and bioinformatics. Nevertheless, for the sake of completeness we have compared our node-based alternative from Table 7 in the main paper to the two most recent methods listed by the Reviewer (neural sheaf diffusion [1] and GRAFF [2]).

We use the two heterophilous datasets that we have in common with the two papers, SQUIRREL and CHAMELEON, with the important observation that in our paper we used the filtered versions recently introduced by Platonov et al. [3]. It was shown in that paper that previous versions contained a data leak leading to unrealistically high performance on heterophilous datasets. Here, we used the best configurations found on the original datasets (from the official repository of each method) and report the metrics against our existing results (Table 1).

The results indicate that even without using our edge-based learning algorithm and attention-based pooling (a PMA layer), the model architecture is expressive enough to outperform these two alternative strategies.

Table 1: MCC for heterophilous datasets, presented as mean \pm standard deviation over 5 runs.

Dataset (\uparrow)	Sheaf diff. [1]	GRAFF [2]	NSA
SQUIRREL	0.23 \pm 0.03	0.22 \pm 0.03	0.29 \pm 0.01
CHAMELEON	0.26 \pm 0.02	0.34 \pm 0.03	0.39 \pm 0.02

Comment 3 (citations for remark)

In the section "Related Work", referring to some transformer approaches, the authors claim that such approaches "did not convincingly outperform simple GNNs". Are there any references to support this claim?

Response

Our remark is derived from several recently published papers. For example from Table 2 in Dwivedi et al. [4] (reference 38 in our manuscript), where the proposed graph transformer did not outperform GatedGCN. Similarly, the method proposed by Shi et al. [5] (reference 37 in our paper) produced improvements over the best GNN in the range of 0.5% to 2% (Tables 4 to 6 in [5]). A recent analysis by Tönshoff et al. [6] also showed that models as simple as GCN or GIN are competitive with or can even outperform sophisticated graph transformers.

Comment 4 (rationale behind Table 8)

In Table 8, what was the heuristic that led the authors to choose such configurations?

Response

The heuristic emerged from a preliminary analysis using a few representative datasets. For instance, we captured the best performing configuration (first row) in Table 8, as well as simple alternations of masked (M) and standard self-attention (S) layers, such as S M S M ... or M S M S ... The latter architectures demonstrate that a simple alternation of layers is often suboptimal. The other architectures are derived from the best configuration, or are meant to capture possibly interesting alternation patterns (e.g. S S M M S S).

Typically, a pattern we observed is that the optimal architectures tend to have self-attention layers at the front and back, masked layers in the middle, and 1 or 2 self-attention layers after the pooling module. These types of architectures could be a good starting point for hyperparameter searches for most tasks.

We have clarified our observations and rationale in the revised manuscript.

Comment 5 (code availability)

The most serious issue, in my opinion, is the lack of availability of code. Even if the authors extensively described the algorithms, this lack keeps the reviewers from having a complete view of the work presented in this manuscript.

Response

We apologise for any miscommunication on our part, as the full code was included in the submission to the journal, including documentation, examples, and set-up instructions, as clarified above. The code will also be available in our revised submission and shortly on GitHub.

Reviewer 2

Comment 1 (edge-centric representation)

The decision to treat graphs as edge sets rather than node-centric structures is interesting but somewhat unconventional. Could the authors further justify why an edge-centric representation is fundamentally superior? Are there theoretical guarantees that ESA can universally capture graph structure better than node-based GNNs and traditional graph transformer methods?

Response

Edge set attention functions on what are termed line graphs in graph theory [7]. In this transformation, edges from the input graph are mapped into nodes in the corresponding line graph, with connections established between them if they share a common node in the original input graph. Prior results have shown that if a pair of graphs is isomorphic then their line graphs are also isomorphic, but the converse does not hold [8, 9]. There are instances where a pair of non-isomorphic graphs is indistinguishable by the 1-WL test, while their line graph counterparts have different 1-WL embeddings. More specifically, take the two graphs from Figure 2 in Wang et al. [10]. There are 7 edges in each of them and the line graphs will, thus, have 7 nodes (each edge from the original graph becomes a node in the corresponding line graph). There is now a node in the second line graph with degree 4 (the bridge edge) vs maximal degree of 3 in the line graph corresponding to the first graph. Hence, 1-WL embeddings for the two line graphs are different.

Another aspect of interest is that in line graphs the focus is on communication of edge tokens that capture information across an edge and its adjacent node features, which is information richer than plain node tokens. Line graphs are claw-free and also tend to highlight specific motifs in the original graph such as triangles, cycles, or cliques [7, 8]. Motifs such as cycles and rings can be important for molecular property prediction tasks. For instance, Bagga and Beineke [11] give a digraph with no cycles that is mapped to isolated vertices via an iterated line graph operator. In contrast to this, a digraph with a pair of joined cycles increases in length under an iterated line graph transformation. While these are examples in the limit, they still provide insights into aspects highlighted by line graph transformations.

When it comes to vertex degrees, a line graph tends to balance the variance relative to the input graph. For instance, take a single node with large degree connected to many other nodes with relatively small degrees. In the corresponding line graph, this would result in a large number of nodes with large vertex degrees and would make the message passing a bit more regular.

In the context of past work on graph neural networks, edge-based message passing has been explored in [12], where it has been hypothesized to help with the noisy nature of node-to-node propagation for certain path types.

Comment 2 (complexity, scalability)

Although ESA utilizes efficient attention computation, it still requires memory quadratic in the number of edges to maintain dense adjacency masks. How does ESA handle extremely large graphs (e.g., graphs with millions of edges)? The paper states that ESA can handle up to 30,000 edges, but is it still scalable beyond this size? Could sparse attention mechanisms (e.g., Performer [1], Longformer-style [2] attention) be further optimized for ESA? Besides, it is recommended to provide a detailed complexity analysis.

Response

Complexity and scalability are characteristics that have shaped our algorithmic design since ESA was first conceptualised. Although we have discussed theoretical aspects of ESA’s complexity in Section 3.3 of the previous version of our manuscript, we agree that several nuances and important scalability observations were left out.

We would first of all like to clarify the statement about handling graphs with 30,000 edges. In this case, we omitted that we were using batches of 8 graphs, and not training on a single graph at a time. Considering that in PyTorch Geometric batching works by diagonally stacking the adjacency

matrices of each individual graph into a single large adjacency matrix, this is equivalent to training on a single graph of around 240,000 edges.

Fundamentally, ESA is enabled by advances in **efficient and exact attention**, as described in the original memory efficient attention paper and several Flash attention papers [13–16]. Efficient attention avoids materialising the $N \times N$ square attention matrix, instead working over smaller chunks and saving partial aggregates that can recover the actual attention scores. The architecture of modern GPUs is used for several optimisations, for example by storing and loading chunks from the GPU’s shared memory/registers, which are much smaller compared to the rest of GPU memory (e.g. HBM), but also much faster. GPUs also come with optimised operations/kernels that are used when memory alignment and tiling constraints are followed (e.g. powers of two).

Thanks to chunking and several optimisations, **exact attention can be computed with linear memory requirements**. Furthermore, although in theory the time complexity is still given by a quadratic number of computations, in practice GPU optimisations such as chunking, fused kernels, reduced memory bottlenecks (better bandwidth utilisation), and others can lead to forward passes that are up to $\times 25$ **times faster**, with comparable uplifts for the backward pass. Ultimately, the exact time and memory statistics depend on the hardware (GPU type), available kernels (for example, CUDA or Triton implementations), and the actual query and key/value chunk sizes that are being used. Notably, the same model can become more efficient by simply upgrading the GPU, since the more powerful hardware and additional components such as tensor cores lead to further optimisations. For example, the transition from Flash attention V2 to V3 on H100 GPUs, which halved the running time.

The complexity of ESA directly corresponds to that of the underlying attention implementation, since that is the main learning mechanism of ESA, augmented with relatively cheap MLPs and normalisation layers.

Indeed, currently the main bottleneck is not the attention computation itself, but storing the dense edge adjacency masks. Unfortunately, all the available libraries that support masking and efficient attention kernels require a dense matrix. This is likely an oversight due to the limited amount of interest in masking, since the main driving force behind efficient attention implementations is scaling up LLM models, where the only masking requirements are simple causal masks or similar pre-defined patterns.

In principle, one only needs to load and apply the masking information per chunk, and in practice this manipulation would lead only to a small overhead, while avoiding hundreds of millions of computations.

It is worth noting the efficiency of these optimised implementations. For a sequence length of 1 million (10^6), an efficient implementation calculates 1 trillion (10^{12}) attention scores. To put this in the context of graph learning and ESA, we generate random Barabasi-Albert (BA) graphs with varying numbers of edges, ranging from 200,000 to 2.6 million. We next train a model using a single self-attention layer (no masking) and representative model parameters (total model dimension 256, 8 attention heads, with MLPs and normalisation layers). Figure 1 illustrates the peak memory use graphically (blue line), demonstrating the linear memory scaling (for the largest graphs, the kernel might be adjusting as the GPU memory limit of 40GB approaches).

We also measured and plotted the amount of memory required to store the **sparse edge masking information**, as opposed to a full dense matrix (green line in Figure 1). Clearly, this cost is relatively trivial even for graphs as large as 2.6 million edges.

We next hypothesise what an efficient implementation might look like by simply adding the sparse storage cost to the self-attention cost (orange line in Figure 1). Note that we are still assuming that full self-attention is being used; however an efficient implementation could avoid hundreds of millions of unnecessary calculations.

We think that this analysis answers the Reviewer’s concerns regarding the potential of ESA in terms of scaling to millions of edges, as well as the use of attention approximations with linear complexity. We consider that the latter are not required, as their use has been largely superseded by exact and efficient attention implementations, which are also widely available and continue to improve. Although the full performance of these operations is still not realised due to the above considerations, we are optimistic that an efficient, bespoke attention implementation for graphs

Figure 1: Memory usage for efficient attention versus the number of edges.

would have a considerable impact on the field. Such an implementation is not trivial, as it requires knowledge of GPU architectures and low-level programming abstractions like Triton and CUDA. As such, it is not within the scope of our current manuscript. Nevertheless, even the current implementation covers a large diversity of application domains, such as computational chemistry (small molecules and peptides), computer vision, bioinformatics, social networks, and more.

We would also like to offer some clarifications regarding the Reviewer’s request for a detailed complexity analysis. In this section, we have established that the complexity of ESA is tied to the underlying attention implementation. Technically, this translates to linear memory complexity and quadratic time complexity (for self-attention layers). However, as discussed above, the theoretical complexity does not align well with the training times seen in the real world due to efficient GPU operations, which can decrease the computation time by more than one order of magnitude compared to standard attention. A precise and detailed complexity analysis would have to include specific GPU architecture details that change from one GPU type to another, as well as a per-case discussion based on the specific query chunk size, key/value chunk size, sequence length (here, number of edges), total hidden dimension, and number of heads. This is because the kernels are available only for certain combinations of sequence lengths, dimensions and heads, and they differ in performance. As such, we think that this is out of the scope of this paper and refer to the Flash attention papers and the corresponding GitHub repository for a more nuanced discussion.

Comment 3 (state-of-the-art performance)

Some state-of-the-art models are missing from comparisons. While ESA is compared against several GNNs and transformers (Graphormer, TokenGT, GPS, etc.), how does ESA compare to Exphormer [3], Polynormer [4], and some pre-training methods for GNNs like GraphMAE [5]?

Response

Given a limited computational budget available to us, the submitted version of the manuscript focused on evaluating a diverse set of baselines and tuning their most important hyperparameters. Our selection of baselines includes the most frequently used methods from the literature

(e.g. GCN, GIN, GraphGPS), strong GNNs that are seldom used in comparisons (PNA), highly expressive models (DropGNN, TokenGT), different attention-based models (GAT, GATv2, and graph transformers), and several instances of the most recent graph transformers (Graphormer and TokenGT which do not use message passing and encode the graph structure through different mechanisms; GraphGPS which is a GNN-Transformer hybrid).

Expormer [17] is a GNN-Transformer hybrid that builds on GraphGPS by adding virtual global nodes and expander graphs. As indicated by the experiments in [17], one should not expect major performance uplifts compared to GraphGPS. For example, it does not appear that there is a statistically significant difference on 3/5 datasets in Table 1 (no t-tests have been done), and similarly in Table 3. Additionally, the experimental results for Polynormer [18] indicate that Expormer and GraphGPS are tied on 3/5 datasets, and we therefore consider GraphGPS as a representative baseline from this class. Moreover, transformations specific to Expormer can also be incorporated into ESA.

We thank the Reviewer for raising our attention to Polynormer [18], which is an interesting combination of graph attention and gating. The approach, however, appears to be evaluated primarily on node classification tasks whereas our main focus is on graph-level predictive performance. We have adapted this baseline to graph-level tasks by including a simple mean pooling layer, and then trained and evaluated tuned Polynormer models.

We note here that training and fine-tuning two additional models on 70 tasks would be computationally expensive, and in addition it might be challenging to formulate a specific hypothesis or insight that these baselines would illustrate, and that has not yet been covered by current baselines over 70 tasks. Hence, we focus our attention on several representative datasets: *i*) relatively large-scale datasets from computational chemistry and computer vision (ZINC and MNIST), *ii*) small-scale molecular datasets (from MOLECULENET), and *iii*) bioinformatics benchmarks (ENZYMES, PROTEINS). The results of our evaluation on these tasks, featuring both Polynormer and Expormer, are provided in Tables 2 to 4. In line with our expectations, Expormer performs similarly to GraphGPS, being better on datasets such as MNIST and PROTEINS, but underperforming slightly on the small molecular datasets and ZINC. Polynormer does not appear to be competitive on the larger graph-level datasets and performs similarly to GraphGPS/Expormer on the smaller benchmarks. The results on heterophilous node classification are more mixed (Table 5); however, even without the edge-based learning algorithm and a pooling (PMA) layer, we achieve state-of-the-art results on the filtered CHAMELEON and SQUIRREL datasets introduced by Platonov et al. [3].

Regarding our empirical results versus leaderboards, at the time of writing, we achieve the highest performance on the ZINC dataset (full version), as compared to 17 other methods, on the ZINC dataset (12K subset version), as compared to 26 other methods, on LRGB-PEPTIDES-STRUCT, as compared to 38 other models, LRGB-PEPTIDES-FUNC, as compared to 43 other models (note that there are two versions, one using the standard protocol of being trained on training data, and one where the validation data is used during training – ESA is the best model in both scenarios), on MNIST without using positional encodings, as compared to 12 other models, and on MALNETTINY, as compared to GraphGPS, Expormer, and tuned GNNs. These results are included in Table 9 in our revised manuscript. Noteworthy is also our result from the paper on PCQM4MV2 without positional encodings, as compared to 19 other models, and where the reported MAE is almost 3 times lower than the next-best model. These leaderboards include results from papers published concurrently or after ESA, and for molecular datasets the previous best results are often achieved by architectures designed specifically for molecules (i.e. not general purpose like ESA, GraphGPS, PNA, etc.).

GraphMAE [19] is an instance of self-supervised learning for graph autoencoders focused on masked feature reconstruction. We think the aim and scope there is substantially different from our approach, and that the work is not directly related to our efforts for effective supervised learning with graphs. We have demonstrated the ability to do effective transfer learning following a recently established strategy [20] and on a new, refined version of the QM9 dataset. In line with that paper and based on our encouraging results from Table 5, it is likely that the benefits apply in general to molecular data, for example in drug discovery campaigns. However, studying transfer learning to this extent is beyond the scope of our current paper. Large-scale pre-training is also highly computationally expensive, suggesting a focused future study.

Table 2: R^2 for MOLECULENET datasets, presented as mean \pm standard deviation over 5 runs.

Dataset (\uparrow)	GPS	Exphormer	Polynormer	ESA
FREESOLV	0.86 \pm 0.03	0.89 \pm 0.01	0.89 \pm 0.01	0.98 \pm 0.00
LIPO	0.79 \pm 0.00	0.75 \pm 0.02	0.80 \pm 0.01	0.81 \pm 0.01
ESOL	0.91 \pm 0.00	0.91 \pm 0.01	0.90 \pm 0.01	0.94 \pm 0.00

Table 3: MCC for classification datasets, presented as mean \pm standard deviation over 5 runs.

Dataset (\uparrow)	GPS	Exphormer	Polynormer	ESA
BBBP	0.705 \pm 0.04	0.675 \pm 0.03	0.736 \pm 0.03	0.835 \pm 0.01
BACE	0.618 \pm 0.03	0.601 \pm 0.02	0.612 \pm 0.04	0.721 \pm 0.02
ENZYMES	0.734 \pm 0.05	0.714 \pm 0.02	0.628 \pm 0.00	0.751 \pm 0.01
PROTEINS	0.443 \pm 0.02	0.520 \pm 0.03	0.572 \pm 0.06	0.589 \pm 0.02
MNIST	0.980 \pm 0.00	0.983 \pm 0.00	0.970 \pm 0.00	0.986 \pm 0.00

Comment 4 (explainability)

Explainability. Given the increasing focus on explainable AI, how interpretable are ESA’s learned representations? Could attention maps be visualized to provide insight into the model’s decision-making process?

Response

Explainability is an aspect where attention models excel compared to other architectures, and one that we did not initially focus on in our manuscript. As ESA is fundamentally edge-based, analysing the attention scores may lead to different insights compared to traditional models.

Here, we investigate two ways of interpreting the learnt attention weights/scores of ESA, inspired by the Polynormer paper [18]. Since ESA excels at graph-level tasks, and especially molecular tasks, we analyse weights from trained ESA models on QM9. This dataset has multiple quantum mechanical prediction targets, enabling the examination of different physical phenomena.

For reference, the extracted attention score tensors have a general shape of `[num_graphs, num_heads, max_edges, max_edges]`, where `num_graphs` is the number of graphs in the dataset, `num_heads` is the number of attention heads, and `max_edges` is the (padded) maximum number of edges in a graph in the dataset. For simplicity, we average on the heads dimensions and analyse the resulting 3D tensor.

Experiment 1 – Distribution of attention scores

We consider a trained QM9 model with the following architecture: `MMMMMSPS` and base our analysis on two properties with significantly different physical characteristics:

1. **HOMO** (highest occupied molecular orbital) energy, an intensive and localised property.
2. **u0** (internal energy at 0K), an extensive and global property.

Since the HOMO can be highly localised on a region of the molecule because it nominally results from a single molecular orbital, while every bond in the molecule contributes to the internal energy and so should be more evenly distributed across all atoms [21, 22], we expect this dynamic to be reflected in the weights learnt by the model. More specifically, for HOMO we hypothesise that only a few attention scores would dominate for each molecule, while for u0 the weights should tend towards the same value.

To quantify this notion, we leverage the Gini coefficient, a measure of inequality that ranges from 0 to 1. A value of 0 indicates equality among all values, while a value of 1 indicates maximal inequality, where a single value dominates and the rest are 0.

Our results are illustrated in Figure 2 for all 7 encoder layers and show that indeed, while the attention scores start from an even spread for both HOMO and u0, in the case of HOMO we notice progressively more peaked and dominant scores until the last layer, while for u0 the opposite is true, with the latter layers exhibiting the most spread out attention.

Table 4: MAE on the ZINC dataset, presented as mean \pm standard deviation over 5 runs.

Dataset (\downarrow)	GPS	Expformer	Polynormer	ESA	ESA (PE)
ZINC	0.024 \pm 0.01	0.041 \pm 0.01	0.101 \pm 0.00	0.027 \pm 0.00	0.015 \pm 0.00

Table 5: MCC for heterophilous datasets, presented as mean \pm standard deviation over 5 runs.

Dataset (\uparrow)	Polynormer	NSA
ROMAN EMPIRES	0.92 \pm 0.00	0.87 \pm 0.00
AMAZON RATINGS	0.38 \pm 0.00	0.34 \pm 0.01
MINESWEEPER	0.77 \pm 0.02	0.69 \pm 0.00
TOLOKERS	0.41 \pm 0.02	0.43 \pm 0.00
SQUIRREL	0.23 \pm 0.02	0.29 \pm 0.01
CHAMELEON	0.31 \pm 0.07	0.39 \pm 0.02

Figure 2: Gini coefficient for attention scores on two quantum properties, HOMO and u0.

Experiment 2 – Top weight bonds/atoms

In the case of the localised HOMO, we investigated if the bonds corresponding to the top attention scores contribute to the “ground truth” molecular orbitals that we computed using Psi4 and the same methodology as in the original QM9 paper (B3LYP functional with the 6-31G(d) basis set). The “ground truth” is in this case, however, an approximation given by visualisation parameters such as the isosurface value. A lower isosurface value includes regions of lower electron density and thus a more extended/diffuse surface, while higher values would highlight only the regions with the highest density. Here, we use the default parameters in Avogadro 2.

It is worth noting that the models are trained on “2D” graph representations without 3D information, and only to predict the scalar HOMO energy. As such, it would be unreasonable to expect high-quality 3D molecular orbital reconstructions. Furthermore, attention scores should be regarded as a proxy and potential explainability tool, but it is difficult if not impossible to establish a strong and reliable relationship between the actual physical coefficients and the model weights. In fact, previous work has shown that physically based models that use the actual orbital coefficients underperform on HOMO energy prediction compared to an attention-based model [21].

We visualise the top attention weights and the HOMO orbitals in Figure 3 for several molecules and notice that the bonds and underlying atoms highlighted by attention are contained within the HOMO 3D regions. The visualisation includes molecules with varied structures and sizes. It is interesting to note that for gdb_39511 (Figure 3C), the molecule is symmetric as far as the 2D model is concerned, and two pairs are highlighted with equal scores. The actual 3D geometry has a different structure, but the highlighted bond corresponds to the visualised orbital.

Comment 5 (sparseness and noise)

ESA focuses on edge-centric representations. How does ESA perform on graphs where edges are sparse or noisy?

Figure 3: Top attention weights illustrated on 2D molecular graphs and HOMO visualisations on the actual 3D geometry.

Response

We identify two aspects of this remark: noisiness and sparsity, which we treat separately.

Regarding noisy graphs, we suspect that the comment might be referring to graph misspecifications, such as “*The encoder vertically interleaves masked and vanilla self-attention modules to learn*

an effective representation of edges, while allowing for tackling possible misspecifications in input graphs” in our abstract. The rationale behind this claim is that self-attention layers without masking are free to attribute attention scores to edges regardless of the graph connectivity, leveraging relationships that would otherwise be missed. For example, in molecular property prediction it is not the bond connectivity that will solely determine the property of a molecule relative to a target protein. In particular, affinity is determined mostly by the 3D arrangement of charged groups in the molecule, and the bond connectivity determines the conformational landscape and hence the possible 3D shapes it can assume. The relationship between shape and bond connectivity is, however, very complex and non-linear. The information about the shape is not specified in the molecular graph that captures bond connectivity, and parts of the molecule that are close in 3D space are not necessarily connected by bonds (exemplified with loops in antibodies, ligand-protein binding, etc.). In this context, masked attention allows for learning effective token representations originating from bond connectivity and combining with self-attention vertically allows for expanding on this information with a strong prior.

More broadly, the notion of “graph noisiness” itself is not well-defined, as it can affect nodes, edges, labels, or other measures, and it could have different implications for different types of data and architectures. For example, randomly dropping nodes can be interpreted as a form of noise, but it has been found to increase the expressiveness of graph neural networks [23]. Moreover, a spurious atom or bond in a molecule does not make physical sense and as such will never be encountered in real-world datasets. At the same time, although “noise” might be present in the data, as far as we know there is no well-known and consistent measure that we could apply to quantify the noise present in the data, which would also be directly relevant to ESA.

With respect to sparseness, we record the number of edges for each graph (E) in a dataset and compare it to the number of edges of a complete graph with the same number of nodes (E_{complete}). More formally, the sparsity factor is given by $1 - \frac{E}{E_{\text{complete}}}$. A graph that approaches E_{complete} can be called “dense” (the number of edges is approximately quadratic compared to the number of nodes). We refer to a graph where the number of edges is close to the number of nodes as “sparse”.

We then averaged the sparseness over all graphs in a dataset and plotted the resulting histogram of the average sparsity per dataset in Figure 4 (we include QM9 and DOCKSTRING only once). The majority of datasets tends to contain sparse graphs (sparsity ≥ 0.9), a prime example being molecular data. However, we encounter a few datasets with slightly lower sparsity (between 0.8 and 0.9), and two instances of dense datasets (the IMDB datasets consisting of ego networks). These cases occur since the graphs have a small number of nodes but are highly connected. Generally, the larger the number of nodes, the less likely it is for the graph to be dense (the heterophily datasets are perfect examples). ESA outperforms GNNs and graph transformers on the dense IMDB datasets, and on the other examples of less sparse data (e.g. PROTEINS and ENZYMES), as can be seen in Table 6 in our manuscript.

Comment 6 (hyperparameters)

How sensitive is ESA to hyperparameter choices (e.g., number of layers, embedding dimension, attention heads)?

Response

Across the 70 datasets and benchmarks used for the evaluation of ESA, we did not observe a particular sensitivity to any hyperparameter. Naturally, different choices of hyperparameters such as the number of layers, hidden dimensions, and attention heads will lead to different performance profiles on different datasets, and this is expected for any neural network model, not just ESA. We have generally found that the explored hyperparameter space discussed in SI 7.1 of our manuscript tends to produce close results, with no cases of divergence or extremely poor models.

ESA allows for any order of masked (M) and self-attention (S) layers in the encoder, as well as for possible inclusion of self-attention layers that follow the pooling layer (P). This is a level of architectural freedom that is, to the best of our knowledge, not present in other graph learning frameworks. While this expands the hyperparameter search space, possibly increasing the computational costs associated with hyperparameter optimisation, the upside is in finding architectures

Figure 4: Histogram of the average graph sparsity per dataset.

with good inductive biases for the target dataset.

We cover this aspect in detail in Section 4.4 of our existing manuscript, where we provide experimental results (Table 8 in the revised version) for varying the layer order and type for 3 datasets from different domains (molecular, computer vision, and heterophilous node classification). Again, while there is a notable difference in performance between different architectures, none of them performed particularly poorly. In addition, a pattern emerges, where most often the optimal architectures tend to have self-attention layers at the front and back, masked layers in the middle, and 1 or 2 self-attention layers after the pooling layer. These types of architectures could be a good starting point for hyperparameter searches for most datasets.

Comment 7 (acronyms)

Many acronyms such as ESA, MAB, SAB, PMA, NSA, are introduced quickly. It is recommended to provide clear definitions when they first appear.

We have ensured that all acronyms are defined when they first appear.

References

1. Bodnar, C., Giovanni, F. D., Chamberlain, B. P., Lio, P. & Bronstein, M. M. *Neural Sheaf Diffusion: A Topological Perspective on Heterophily and Oversmoothing in GNNs* in *Advances in Neural Information Processing Systems* (2022).
2. Giovanni, F. D., Rowbottom, J., Chamberlain, B. P., Markovich, T. & Bronstein, M. M. Understanding convolution on graphs via energies. *arXiv/2206.10991* (2023).
3. Platonov, O., Kuznedelev, D., Diskin, M., Babenko, A. & Prokhorenkova, L. *A critical look at the evaluation of GNNs under heterophily: Are we really making progress?* in *The Eleventh International Conference on Learning Representations* (2023).
4. Dwivedi, V. P. & Bresson, X. A Generalization of Transformer Networks to Graphs. *AAAI Workshop on Deep Learning on Graphs: Methods and Applications* (2021).
5. Shi, Y. *et al.* *Masked Label Prediction: Unified Message Passing Model for Semi-Supervised Classification* in *Proceedings of the Thirtieth International Joint Conference on Artificial Intelligence* (2021).

6. Tönshoff, J., Ritzert, M., Rosenbluth, E. & Grohe, M. *Where Did the Gap Go? Reassessing the Long-Range Graph Benchmark* in *The Second Learning on Graphs Conference* (2023).
7. Harary, F. & Norman, R. Z. Some properties of line digraphs. *Rendiconti del Circolo Matematico di Palermo* **9**, 161–168 (1960).
8. Whitney, H. Congruent Graphs and the Connectivity of Graphs. *American Journal of Mathematics* **54**, 150–168 (1932).
9. Bauer, D. & Tindell, R. Graphs isomorphic to subgraphs of their line-graphs. *Discrete Mathematics* **41**, 1–6 (1982).
10. Wang, Q., Chen, D. Z., Wijesinghe, A., Li, S. & Farhan, M. *\mathcal{N} -WL: A New Hierarchy of Expressivity for Graph Neural Networks* in *The Eleventh International Conference on Learning Representations* (2023).
11. Bagga, J. & Beineke, L. A survey of line digraphs and generalizations *. *Discrete Mathematics Letters* **6** (Mar. 2021).
12. Heid, E. *et al.* Chemprop: A Machine Learning Package for Chemical Property Prediction. *Journal of Chemical Information and Modeling* **64**, 9–17 (2024).
13. Rabe, M. N. & Staats, C. Self-attention Does Not Need $O(n^2)$ Memory. *arXiv/2112.05682* (2021).
14. Dao, T., Fu, D. Y., Ermon, S., Rudra, A. & Ré, C. *FlashAttention: Fast and Memory-Efficient Exact Attention with IO-Awareness* in *Advances in Neural Information Processing Systems* (2022).
15. Dao, T. *FlashAttention-2: Faster Attention with Better Parallelism and Work Partitioning* in *International Conference on Learning Representations (ICLR)* (2024).
16. Shah, J. *et al.* FlashAttention-3: Fast and Accurate Attention with Asynchrony and Low-precision. *arXiv/2407.08608* (2024).
17. Shirzad, H., Velingker, A., Venkatachalam, B., Sutherland, D. J. & Sinop, A. K. *Expformer: Sparse transformers for graphs* in *International Conference on Machine Learning* (2023).
18. Deng, C., Yue, Z. & Zhang, Z. *Polynormer: Polynomial-Expressive Graph Transformer in Linear Time* in *The Twelfth International Conference on Learning Representations* (2024).
19. Hou, Z. *et al.* *GraphMAE: Self-Supervised Masked Graph Autoencoders* in *Proceedings of the 28th ACM SIGKDD Conference on Knowledge Discovery and Data Mining* (2022).
20. Buterez, D., Janet, J. P., Kiddle, S. J., Oglic, D. & Lió, P. Transfer learning with graph neural networks for improved molecular property prediction in the multi-fidelity setting. *Nature Communications* **15**, 1517. ISSN: 2041-1723 (Feb. 2024).
21. Buterez, D., Janet, J. P., Kiddle, S. J., Oglic, D. & Lió, P. Modelling local and general quantum mechanical properties with attention-based pooling. *Communications Chemistry* **6**, 262. ISSN: 2399-3669 (Nov. 2023).
22. Chen, K., Kunkel, C., Cheng, B., Reuter, K. & Margraf, J. T. Physics-inspired machine learning of localized intensive properties. *Chem. Sci.* **14**, 4913–4922 (18 2023).
23. Papp, P. A., Martinkus, K., Faber, L. & Wattenhofer, R. DropGNN: Random Dropouts Increase the Expressiveness of Graph Neural Networks. *arXiv/2111.06283* (2021).

An end-to-end attention-based approach for learning on graphs

Point-by-point response

We would like to once again thank the Editor and the Reviewers for their time, feedback, and constructive comments. For our latest revision, we did not receive any additional requests or comments from the Reviewers, other than stating that all concerns were addressed.

Review of paper "An end-to-end attention-based approach for learning on graphs"

The present work proposes a new graph learning model, designed on a purely attention-based approach that work on graphs seen as set of edges, and a thorough comparison with many popular GNNs has been conducted in terms of accuracy, time and memory scaling. The manuscript is well-written, the model design is convincing, also given the extensive numerical study conducted by the authors.

I have some general observations about the manuscript:

- there are some typos in the manuscript (e.g., end of page 2, "Attention coefficients **there** are defined..."); I would suggest to revise it from a grammar and syntax point of view;
- in the "Introduction" section the authors talk about the well-known issues of oversmoothing and oversquashing; nevertheless, in my opinion very important references in literature concerning these concepts are not addressed, e.g. [1, 2, 3]; in general, there is a lack of comparison with these energy-based graph learning models;
- In the section "Related Work", referring to some transformer approaches, the authors claim that such approaches "did not convincingly outperform simple GNNs". Are there any references to support this claim?
- In Table 8, what was the heuristic that led the authors to choose such configurations?
- the most serious issue, in my opinion, is the lack of availability of code. Even if the authors extensively described the algorithms, this lack keeps the reviewers from having a complete view of the work presented in this manuscript

Once the authors address the abovementioned issues, I could recommend the paper for publication.

References

- [1] C. Bodnar, F. Di Giovanni, B. Chamberlain, P. Lio, and M. Bronstein. Neural sheaf diffusion: A topological perspective on heterophily and oversmoothing in gnns. *Advances in Neural Information Processing Systems*, 35:18527–18541, 2022.
- [2] F. Di Giovanni, J. Rowbottom, B. P. Chamberlain, T. Markovich, and M. M. Bronstein. Graph neural networks as gradient flows. 2022.
- [3] J. Topping, F. Di Giovanni, B. P. Chamberlain, X. Dong, and M. M. Bronstein. Understanding over-squashing and bottlenecks on graphs via curvature. In *International Conference on Learning Representations*.